# Mass balance, grade, and adjustment timescales in bedrock channels

Jens Martin Turowski[1]

[1] Helmholtzzentrum Potsdam, German Research Centre for Geosciences GFZ, Telegrafenberg, 14473 Potsdam, Germany

*Correspondence to*: Jens M. Turowski (turowski@gfz-potsdam.de)

**Abstract**

Rivers are dynamical systems that are thought to evolve towards a steady state configuration. Then, geomorphic parameters, such as channel width and slope, are constant over time. In the mathematical description of the system, the steady state corresponds to a fixed point in the dynamic equations in which all time derivatives are equal to zero. In alluvial rivers, steady state is characterised by grade. This can be expressed as a so-called order principle: an alluvial river evolves to achieve a state

in which sediment transport is constant along the river channel, and is equal to transport capacity everywhere. In bedrock rivers, steady state is thought to be achieved with a balance between channel incision and uplift. The corresponding order principle is: A bedrock river evolves to achieve a vertical bedrock incision rate that is equal to the uplift rate or baselevel lowering rate. In the present work, considerations of process physics and of the mass balance of a bedrock channel are used to argue that bedrock rivers evolve to achieve both grade and a balance between channel incision and uplift. As such, bedrock

channels are governed by two order principles. As a consequence, the recognition of a steady state with respect to one of them does not necessarily imply an overall steady state. For further discussion of the bedrock channel evolution towards a steady state, expressions for adjustment timescales are sought. For this, a mechanistic model for lateral erosion of bedrock channels is developed, which allows to obtain analytical solutions for the adjustment timescales for the morphological variables of channel width, channel bed slope and alluvial bed cover. The adjustment timescale to achieve steady cover is of the order of

minutes to days, while the adjustment timescales for width and slope are of the order of thousands of years. Thus, cover is adjusted quickly in response to a change in boundary conditions to achieve a graded state. The resulting change in vertical and lateral incision rates triggers a slow adjustment of width and slope, which in turn affects bed cover. As a result of these feedbacks, it can be expected that a bedrock channel is close to a graded state most of the time, even when it is transiently adjusting its bedrock channel morphology.

**1 Introduction**

Bedrock rivers are important geomorphic landforms in mountain regions. They set the baselevel for hillslope response and evacuate the products produced by erosion, weathering and hillslope mass wasting (e.g., Hovius and Stark, 2006). As such, they integrate the upstream erosional signal of the landscape, and the material transported in rivers can be used to estimate catchment-averaged denudation rates on various timescales (e.g., Turowski and Cook, 2017). Further, their morphology is

30 thought to be indicative of past climate and tectonic conditions (e.g., Stark et al., 2010; Wobus et al., 2006). Consequently, they provide archives that can be exploited to unravel the Earth's history.

River channels are dynamical systems. Their state variables – for example, slope, cross-sectional shape, and bed roughness – evolve over time under the influence of externally imposed driving variables including water discharge, sediment supply, and

35 tectonic uplift (e.g., Heimann et al., 2015; Lague, 2010; Parker, 1979; Wickert and Schildgen, 2019). Like in many other dynamical systems, there exists a fixed point in the descriptions of river dynamics, at which all state variables are constant over time. In an alluvial river, at this fixed point, entrainment and deposition of sediment are in balance along the river profile, implying that sediment transport rate is constant and that sediment transport capacity matches sediment supply. A river that exhibits these features is said to be 'in grade' or 'graded', because it is neither aggrading nor degrading (Mackin, 1948). Since

its introduction, the graded stream concept has become a central paradigm in river morphodynamics (e.g., Blom et al., 2017; Church, 2006). There are several reasons for this importance. Chiefly, rivers are physically complicated systems, and the description of their steady state forms is a problem that is considerably simpler than the full description of their dynamics. Further, many variables of natural rivers are challenging to measure. Yet, comparatively simple scaling relations have been

observed between variables such as discharge or drainage area, on the one hand, and channel width or channel slope, on the other hand (e.g., Gleason, 2015; Leopold and Maddock, 1953; Whitbread et al., 2015). These scaling relationships are thought to be explainable using steady state models (e.g., Eaton and Church, 2004; Smith, 1974; Turowski, 2018; Wobus et al., 2006).

The condition of grade in a stream is tightly connected to the description of its sediment mass balance. For alluvial rivers, this

mass balance is typically described by one of two approaches, the Exner equation or the entrainment-deposition framework (e.g., An et al., 2018). In the Exner equation (e.g., Chen et al., 2014; Paola and Voller, 2005), the rate of change of the sediment bed elevation $h_s$ is related to the long-stream divergence of sediment supply per unit width, $q_s$.

$$\frac{\partial h_s}{\partial t} = -\frac{1}{\rho_s(1-p)}\frac{\partial q_s}{\partial x}$$

(1)

Here, $p$ is the porosity and $\rho_r$ the density of the sediment, $t$ the time and $x$ the distance in the downstream direction. In steady state, for a graded stream, the time derivative on the left-hand side is zero, which implies that the spatial derivative on the right-hand side is zero also. As a result, the sediment flux is constant along the stream – the stream is in grade. Any bed elevation change leads to an adjustment of slope. The condition of grade thus implies that transport capacity is also constant and equal to sediment supply along the stream. In the entrainment-deposition framework (e.g., Charru et al., 2004; Davy and

Lague, 2009; Shobe et al., 2017), the entrainment rate $E$ and deposition rate $D$ of sediment mass per unit area are tracked explicitly, giving the mass balance for the mobile sediment mass per unit area $M_m$ (e.g., Turowski and Hodge, 2017)

$$\frac{\partial M_m}{\partial t} = -\frac{\partial q_s}{\partial x} + E - D$$

(2)

The sediment bed elevation change is then described by a second equation

$$\frac{\partial h_s}{\partial t} = \frac{1}{\rho_s(1-p)}(D-E)$$

(3)

Within this framework, in steady state, time derivatives are set to zero, implying that entrainment needs to equal deposition (eq. 3) and sediment flux along the stream needs to be constant (eq. 2). Again, this means that the stream is in grade. The main advantage of the erosion-deposition framework is that it keeps separately track of stationary and moving sediment mass. This

is allows to predict a lagged response of bed elevation to changes in sediment supply, due to the interplay of entrainment, deposition and lateral sediment movement (e.g., An et al., 2018). Its main disadvantage is that both entrainment and deposition ($E$ and $D$ in eq. 2 and 3) need to be quantified in terms of hydraulic drivers. In contrast, to use the Exner equation, only transport capacity or transport rate needs to be quantified, which is considerably easier to measure than deposition and entrainment rate, and therefore the relevant relationships are better constrained. Nevertheless, both approaches are related and the entrainment-

deposition equations (2 and 3) can be transformed into the Exner equation (1) when combining mobile and stationary mass into a single total mass term (Appendix A).

In bedrock channels, the concept of grade has not been widely applied. One of the main reasons for this is that bedrock channels are usually viewed as detachment-limited systems, where sediment supply is much smaller than transport capacity (e.g., Tinkler

and Wohl, 1998; Whipple et al., 2013), which is in direct contrast to the assumption of grade. As a result, the system is assumed

to be driven by its potency for erosion (e.g., Whipple, 2004). The evolution of bedrock channel bed elevation $h_b$ is described by the equation (e.g., Howard, 1994)

$$\frac{\partial h_b}{\partial t} = T_U - I$$

(4)

Here, $T_U$ is the uplift rate or relative baselevel fall rate and $I$ the bedrock incision rate. According to equation (4), bedrock channels adjust to a steady state in which incision rate $I$ equals uplift rate $T_U$.

Over the last two decades, evidence has been mounting that fluvial bedrock erosion is driven by the impacts of sediment particles in many settings (e.g., Cook et al., 2013; Johnson et al., 2010; Sklar and Dietrich, 2004). The amount of sediment in
the channel affects erosion rates by two main effects. First, an increase in the number of moving particles leads to an increase in the number of impacts on the bed, increasing erosion rates. This is known as the tools effect. Second, sediment residing on the bed may protect the rock surface from impacts, reducing erosion rates. This is known as the cover effect. Evidence for both tools and cover effects have been described in laboratory and field studies (e.g., Beer et al., 2016; Cook et al., 2013; Johnson and Whipple, 2010; Sklar and Dietrich, 2001; Turowski et al., 2008a). In addition, large sediment bodies are common in many
mountain regions can reside in mountain areas in and around stream channels for potentially long time (e.g., Korup et al., 2006; Schoch et al., 2018). All of these observations imply that a description of the mass balance of sediment should be an essential part of any theoretical description of bedrock channels. In addition, recent observations have been interpreted such that bedrock channels are in a graded state, similar to alluvial channels (Phillips and Jerolmack, 2016). Thus, it seems that the view that bedrock channels are in a detachment-limited state, in which long-term sediment supply is smaller than transport
capacity (e.g., Whipple et al., 2013), is insufficient to account for all observations made in natural streams.

In this paper, I have three separate, yet related aims. First, I develop a description of the mass balance of bedrock channels, based on previous work by Turowski and Hodge (2017) and Turowski (2018). The mass balance is used to derive and discuss the concept of the graded stream for bedrock channels. Second, I derive expressions for response time scales for bedrock
channels to adjust to a graded state. Third, for this it is necessary to develop a description of bedrock channel wall erosion by impacting particles. The concepts are used to discuss the current notion of bedrock channels, their possible routes to a graded state and the relevant response time scales.

## 2 Theoretical considerations

### 2.1 Mass balance equations for sediment

Landscapes form by the interplay of bedrock erosion, and the entrainment, transport and deposition of sediment, as determined by various drivers such as climate, tectonics, and biological activity. Each erosion process has a minimum of two phases: the break-down of rock mass by chemical or physical weathering, and the entrainment and evacuation of loose pieces of rock that are produced in this way (Gilbert, 1877). From this, it is clear that a minimum description of any eroding landscape needs to include a mass balance equation each for bedrock and for loose sediment. Consider a control volume within a river (Fig. 1),
with width $W$, length $L$, and a height ranging from the surface, i.e., the interface between bedrock or sediment and the atmosphere, to a fixed reference level somewhere in the bedrock below. The loose material, sediment, overlays the bedrock. Uplift pushes new bedrock into the control volume at a rate $T_U$, while incision converts it into sediment at a rate $I$. We assume that the erosion products are small enough so that they are subsequently transported in suspension. Then the rate of change of bedrock mass per unit area $M_b$ is given by:

$$\frac{\partial M_b}{\partial t} = \rho_r(T_U - I)$$

(5)

Here, $\rho_r$ the density of the bedrock. Dividing eq. (5) by $\rho_r$, and realizing that $h_b = M_b/\rho_r$, we retrieve the usual form of the bedrock mass balance, eq. (4). Details of the derivation of the mass balance for sediment have been given by Turowski and Hodge (2017). Note that working with mass instead of a deposit thickness is advantageous for bedrock channels, because sediment may not be equally distributed on the bed. The entrainment-deposition framework is preferable, because it makes possible to distinguish between moving and stationary sediment, which is necessary to treat the cover and the tools effects. This is not possible when using the Exner approach (Appendix A). The mass balance for the mobile sediment per unit area $M_m$ is given by equation (2)

$$\frac{\partial M_m}{\partial t} = -\frac{\partial q_s}{\partial x} + E - D$$

(6)

The mass balance for the stationary sediment per unit area $M_s$ is given by

$$\frac{\partial M_s}{\partial t} = D - E$$

(7)

Finally, sediment flux $q_s$ and mass $M_m$ are connected via the downstream particle speed $U$:

$$q_s = UM_m$$

(8)

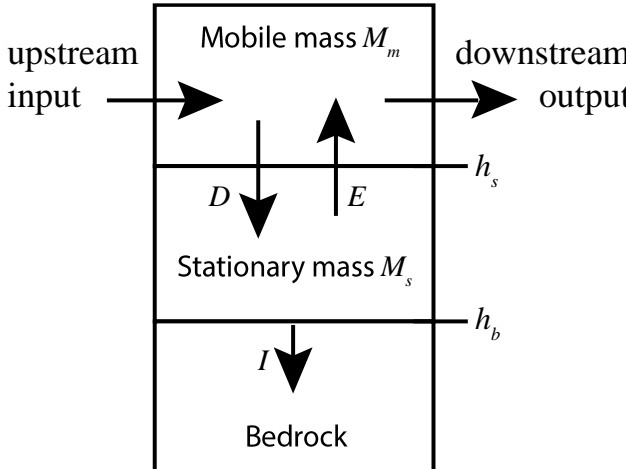

**Figure 1: Schematic side view of a control volume within a bedrock channel. The bedrock (bottom) is overlain by stationary sediment (centre), which exchanges particles via entrainment $E$ and deposition $D$ with the mobile sediment in the water column (top). The bedrock surface $h_b$ lowers at the incision rate $I$, while the sediment surface $h_s$ evolves according to the balance of entrainment and deposition (eq. 6).**

**2.2 Lateral erosion in bedrock channels by impacting particles**

Considering impact erosion to be the dominant erosion process, the lateral erosion rate $E_L$ of bedrock channels is driven by particle impacts. It can therefore, similar to the formulation of the saltation-abrasion model (Sklar and Dietrich, 2004), be written as the product of two terms: (i) the average volume eroded by a single impact, $V_i$, and (ii) the impact rate per area and time $I_r$. The latter term can be subdivided into two terms. The first of these quantifies the number of available particles per unit time and area, $F_T$, which describes the tools effect. The second term $F_C$ describes the effect of bed cover, which captures the effects of the distribution of sediment in the channel on lateral erosion. The need for this term arises because bedload particles

generally travel parallel to the channel walls. Sideward deflection is controlled by the interaction of moving particles with the bed (Beer et al., 2017; Fuller et al., 2016), and specifically with stationary sediment, i.e., bed cover (cf. Turowski, 2018).

$$E_L = V_i I_r = V_i F_T F_C$$

(9)

The volume eroded per impact for lateral erosion should be the same as for vertical erosion and has been quantified by Sklar and Dietrich (2004) as the energy of the impact divided by a material constant. It can be evaluated by

$$V_i = \frac{2Y}{k_v \sigma_T^2} \frac{M_p w_i^2}{2}$$

(10)

Here, the first term is related to material properties, where $Y$ and $\sigma_T$ are Young's modulus of the bedrock and its tensile strength, respectively, and $k_v$ is the rock resistance coefficient. The second term gives the kinetic energy of the impacting grain. Here, $M_p$ is the mass of a single particle and $w_i$ the impact speed normal to the wall.

As in vertical bedrock erosion (Beer and Turowski, 2015; Inoue et al., 2014; Sklar and Dietrich, 2004), the tools effect can be modelled as a linear function of bedload supply $Q_s$ (Mishra et al., 2018), multiplied by a dimensionless factor $\kappa_T$ with values between 0 and 1 that describes the fraction of bedload available for lateral erosion. To obtain the number of impacting particles per unit area, this product needs to be divided by the mass of a single particle and the total area of the wall $A_w$ that is eroded, to give

$$F_T = \frac{\kappa_T Q_s}{A_w M_p}$$

(11)

Substituting eqs. (10) and (11) into (9), the lateral erosion rate of a bedrock channel can thus be written as

$$E_L = \kappa_T \frac{Y}{k_v \sigma_T^2} \frac{Q_s w_i^2}{A_w} F_{CD}$$

(12)

In eq. (12), there are three parameters that require further discussion: the impact speed $w_i$, the eroded area $A_w$ and the cover-dependent term $F_{CD}$. In a previous paper (Turowski, 2018), I argued that lateral erosion and channel width development are intimately related to bed cover. The quantification of all three parameters springs from the physical-conceptual model developed in this previous paper. For this reason the cover-dependent term, $F_{CD}$, will be discussed first, leading to a quantification of the other two terms, $w_i$ and $A_w$.

In a straight bedrock channel the motion of water and sediment is generally parallel to the walls. Lateral erosion occurs when sediment particles are deflected sideways such that they impact the walls with sufficient force to cause damage. For a given reach, we can define a sideward deflection length scale $d$, which is relevant for reach-scale lateral erosion (Turowski, 2018). The relevant cross section for setting reach-scale channel width is assumed to be located where the sinuous bedload particle stream crosses from the gravel bar onto the smooth bedrock at the apex of the bar (Fig. 2). Only there, several conditions come together that are favourable to achieve the maximal sideward deflection distances (Turowski, 2018). These are (i) the high particle concentration, (ii) a vector of motion of the particle stream that is already pointing towards the walls, (iii) the existence of roughness necessary for sideward deflection provided by the alluvium, and (iv) the smooth bedrock that does not hinder sideward motion. We expect that the wall is eroded if the uncovered width $W_{uncovered}$ in the cross section is smaller than $d$ (Fig. 3). As a result, we can quantify the cover-dependent term $F_{CD}$ as

$$F_{CD} = \begin{cases} 1 & \text{if} \quad d > W_{uncovered} \\ 0 & \text{otherwise} \end{cases}$$

(13)

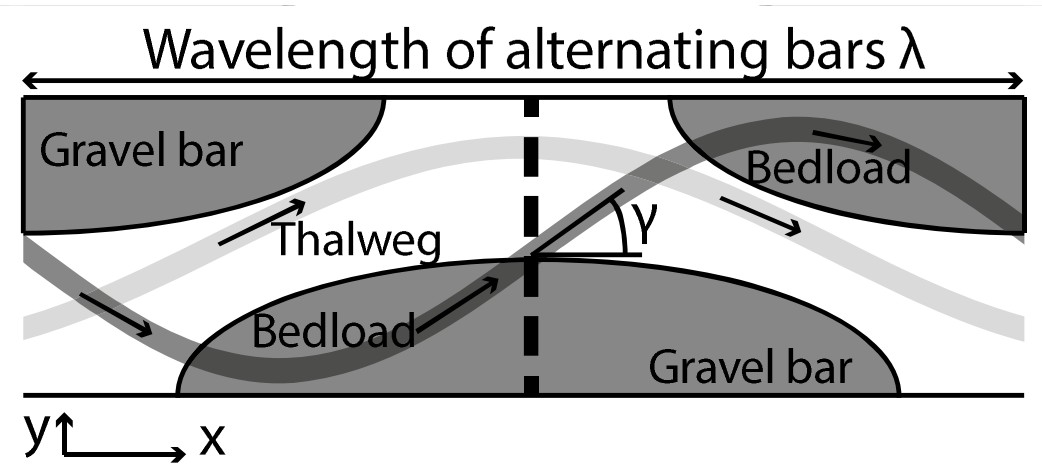

**Figure 2: Schematic top view of a straight bedrock channel, with alternating submerged gravel bars (dark grey) on a bedrock bed (white). The sinuous thalweg (light grey) and bedload path (transparent dark grey) are indicated. The black dashed line indicates the cross section that is ideal for sideward deflection of particles; here, the bedload particle stream crosses the boundary between gravel and smooth bedload. The wavelength of the alternating bars and therefore of the bedload path should scale with channel width. Adapted from Turowski (2018).**

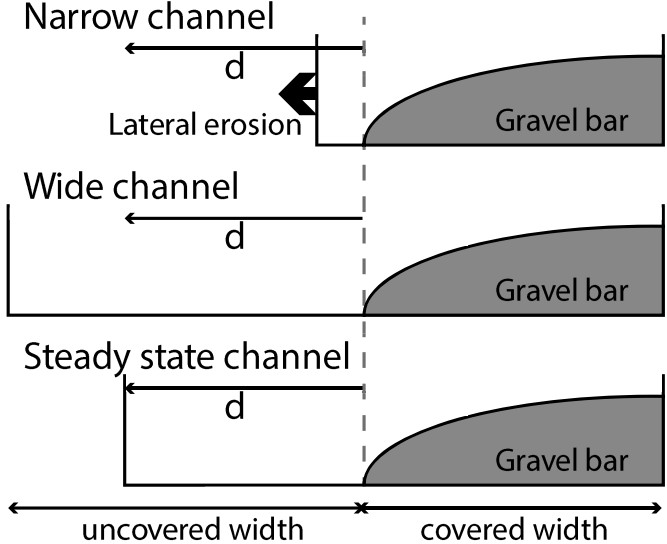

**Figure 3: The sideward deflection length scale _d_ interacts with bed cover and channel width to determine whether the lateral erosion occurs (top), or not (center, bottom). Adapted from Turowski (2018).**

We can write the eroded area on the wall $A_w$ as the product of a length scale and a height. From the argument above, the same particle can attack the wall once when passing each gravel bar. Therefore, the relevant length scale for lateral erosion is the distance between bars on a given side of the channel, i.e., the wavelength of bar spacing, $\lambda$. Fuller et al. (2016) observed that for sideward-deflected particles, the erosion height on the wall is larger than the typical saltation hop height. Beer et al. (2017) observed a similar increase in wall erosion rates near boulder obstacles in the channel. When the roughness elements that cause deflection are related to stationary alluvium, we can expect that the height scale is the maximum saltation height of bedload particles at the wall, $H_w$.

$$A_w = \lambda H_w$$

(14)

Note that not the entire area is eroded at the same time. Rather, particles are deflected towards the wall near the apex of the bars (cf. Turowski, 2018). Consequently, only a small area is eroded at a given time, and the locus of erosion slowly moves downstream as the bars migrate. Likewise, the impact speed $w_i$ and the sideward deflection distance $d$ are related to saltation properties. It is, of course, possible that a particle undergoes several saltation cycles until it impacts the wall. However, in this

case, in each additional saltation hop, the sideward component of motion would reduce due to downstream hydraulic forces and frictional loss of momentum. Here, I assume that only during the first hop, particles have sufficient lateral momentum to cause erosion upon impact on the wall. This assumption needs to be verified experimentally.

Since, within the model, sideward deflection is caused by stationary alluvium, particle trajectories should follow those

observed for saltation over alluvium (e.g., Abbot and Francis, 1977; Niño et al., 1994), rather than those over bedrock (e.g., Chatanantavet et al., 2013; Auel et al., 2017a). Because the wall-normal component of the motion is relevant for impact erosion (e.g., Sklar and Dietrich, 2004), the particle trajectory needs to be corrected for its angle $\gamma$ of motion with respect to the wall. Then, the sideward deflection distance $d$ is related to the saltation hop length $L_s$ by

$$d = L_s \sin(\gamma)$$

(15)

Likewise, the impact speed $w_i$ is related to the particle speed $U$ by

$$w_i = U \sin(\gamma)$$

(16)

Auel et al. (2017a) proposed empirical equations to describe saltation properties over a sediment bed as a function of

hydraulics, based on their own experiments and a data compilation from various sources. They give the saltation hop length $L_s$ by

$$\frac{L_s}{D} = 1.17 \left( \frac{\theta}{\theta_c} - 1 \right)$$

(17)

Here, $D$ is grain diameter, $\theta$ is the Shields stress and $\theta_c$ the critical Shields stress for the onset of bedload motion. Similarly,

hop height $H_s$ is given by

$$\frac{H_s}{D} = 0.025 \left[ \left( \frac{\theta}{\theta_c} - 1 \right) + 24 \right]$$

(18)

and downstream particle speed $U$

$$U = 1.46 \left[ \left( \frac{\rho_s}{\rho} - 1 \right) g D \right]^{0.5} \left( \frac{\theta}{\theta_c} - 1 \right)^{0.5}$$

(19)

Here, $\rho$ and $\rho_s$ are the densities of the water and sediment, respectively, and $g$ is the acceleration due to gravity. Finally, to close the system of equations, we need some relations describing the geometry of the gravel bars. Alternating bars in bedrock channels have been little studied (e.g., Nelson and Seminara, 2012), and the necessary relations are not available. From a large

data compilation of bar width and length in braided channels, Kelly (2006) found that bar length $L_{bar}$ is related to bar width $W_{bar}$ by

$$L_{bar} = 4.95 W_{bar}^{0.97}$$

(20)

Based on this observation, I assume that in bedrock channels the wavelength of the bars scales with their width, such that

$$\lambda = \kappa_{bar} W_{bar} = \kappa_{bar} W_{covered}$$

(21)

Here, the bar width has been identified with the covered width $W_{covered}$ (Fig. 2, 3), and $\kappa_{bar}$ is a dimensionless constant with a value of 2-10, in analogy with bar shapes in alluvial rivers (e.g., Kelly, 2006). Bed cover $C$ is the ratio of covered bed area $A_c$ to total bed area $A_{tot}$, which can be related to the covered width $W_{covered}$ as follows

$$C = \frac{A_c}{A_{tot}} = \frac{W_{covered}}{W} = \frac{W_{bar}}{W}$$

(22)

Here, $W$ is the channel width. As a result, the bar length can be written as

$$\lambda = \kappa_{bar} W C$$

(23)

Assuming that the maximum saltation height at the wall corresponds to the maximum saltation hop height, $H_w = H_s$, and substituting eqs. (13) to (16), and (18) to (23) into (12), we obtain

$$E_L = \begin{cases} \dfrac{\kappa Y g}{k_v \sigma_T^2}\left(\dfrac{\rho_s}{\rho} - 1\right)\sin^2(\gamma)\dfrac{Q_s}{WC}\dfrac{\left(\dfrac{\theta}{\theta_c} - 1\right)}{\left(\dfrac{\theta}{\theta_c} - 1\right) + 24} & \text{if} \quad d > W_{uncovered} \quad \text{and} \quad \theta > \theta_c \\ 0 & \text{otherwise} \end{cases}$$

(24)

Here, $\kappa = 85\kappa_T/\kappa_{bar}$ is a dimensionless constant. The sideward deflection length scale $d$ can be estimated by the hop length $L_s$ (eq. 24)

$$d = 1.17 D \sin(\gamma)\left(\frac{\theta}{\theta_c} - 1\right)$$

(25)

Finally, the uncovered width can be related to bed cover using eq. (22).

$$W_{uncovered} = W - W_{covered} = W(1 - C)$$

(26)

The rate of change of channel width, in case of a widening channel, should be twice the lateral erosion rate given in eq. (24), since both sides are eroded at the same time.

$$\frac{dW}{dt} = 2E_L$$

(27)

Note that, when $d = W_{uncovered}$, the model gives a stead state channel width consistent with the model of Turowski (2018), with the sideward deflection distance given by eq. (25).

## 2.3 Timescales of morphological adjustment in bedrock channels

I will now derive analytical expressions for the response time of the channel to perturbations in the boundary conditions, such as changes in discharge, sediment supply or uplift rate. This will be done for three key parameters, channel bed slope, channel width, and cover. For the derivation, it is necessary to assume that, on the time scale of adjustment of one variable, the other variables stay essentially constant. This assumption is reasonable, if a particular variable adjusts much slower than another. For example, slope adjustment takes much longer times than the adjustment of bed cover.

### 2.3.1 Response time of channel bed slope

Taking the spatial derivative of eq. (4) and assuming spatially constant uplift rate $T_U$, we obtain

$$\frac{\partial}{\partial x}\frac{\partial h_b}{\partial t} = -\frac{\partial I}{\partial x}$$

(28)

Channel bed slope $S$ is defined as the topographic gradient in the downstream direction

$$S = -\frac{\partial h_b}{\partial x}$$

(29)

Equation (28) can thus be rewritten as

$$\frac{\partial I}{\partial x} = -\frac{\partial}{\partial x}\frac{\partial h_b}{\partial t} = -\frac{\partial}{\partial t}\frac{\partial h_b}{\partial x} = \frac{\partial S}{\partial t}$$

(30)

According to the revised saltation-abrasion equation by Auel et al. (2017b), the vertical erosion equation takes the form

$$I = \frac{gY}{230k_v\sigma_T{}^2}\left(\frac{\rho_s}{\rho}-1\right)\frac{Q_s}{W}(1-C)$$

(31)

Steady state cover can be described with the equation by Turowski and Hodge (2017)

$$C = \left(1 - e^{-\frac{Q_s}{M_0 UW}}\right)\frac{Q_s}{Q_t}$$

(32)

The bedload transport capacity can be written as

$$\frac{Q_t}{W} = K_{bl}Q^m S^n$$

(33)

Substituting eqs. (29) to (33) into (28), and assuming that all variables apart from slope are constant, the slope evolution equation takes the form

$$\frac{\partial S}{\partial t} + nBS^{-n-1}\frac{\partial S}{\partial x} = 0$$

(34)

Here, $B$ is assumed to be constant.

$$B = \frac{gY}{230k_v\sigma_T{}^2}\left(\frac{\rho_s}{\rho}-1\right)\left(1 - e^{-\frac{Q_s}{M_0 WU}}\right)\frac{Q_s{}^2}{K_{bl}W^2 Q^m}$$

(35)

Equation (34) is a non-linear wave equation with celerity $c_S$

$$c_S = nBS^{-n-1}$$

(36)

The time scale of slope adjustment $T_S$ can therefore be written as

$$T_S = \frac{L}{c_S} = \frac{LS^{n+1}}{nB} = \frac{k_{bl}Q^m LS^{n+1}}{nk\left(1 - e^{-\frac{q_s}{M_0 U}}\right)q_s{}^2} = \frac{q_t LS}{nk\left(1 - e^{-\frac{q_s}{M_0 U}}\right)q_s{}^2} = \frac{LSW}{nkQ_s C}$$

(37)

Here, $L$ is the length of the reach in question, and $k$ is the erodibility, which, according to the revised saltation-abrasion equation by Auel et al. (2017b) takes the form

$$k = \frac{gY}{230k_v\sigma_T{}^2}\left(\frac{\rho_s}{\rho}-1\right)$$

(38)

### 2.3.2 Response time of channel width

For the adjustment of channel width, it is necessary to distinguish between narrowing and widening channels. While channel widening is controlled by the lateral erosion of bedrock walls (see section 2.2, eq. 24), a bedrock channel can only narrow when incising vertically. Therefore, the response time scale of narrowing is related to the vertical incision rate. The timescale of narrowing can be estimated by the time necessary incise the flow depth $H$. After this time, the wetted channel cross-section has been completely replaced. Thus, using the continuity equation (D3) and the expression for flow velocity (D4), the time scale of channel narrowing is

$$T_N = \frac{H}{I} = \frac{(gS)^{\frac{\alpha-1}{2}} R^{\frac{3\alpha-1}{2}}}{k_V I} \left(\frac{Q}{W}\right)^{1-\alpha}$$

(39)

The technique of perturbation analysis can be used to obtain an analytical solution for the width response time in case of a widening channel (e.g., Braun et al., 2015, Turowski and Hodge, 2017). The mathematical details are given in Appendix C, leading to the equation

$$T_W = \frac{18 k_v \sigma_T^2}{\kappa Y \left(\frac{\rho_s}{\rho}-1\right) g} \frac{\theta_c}{\theta} \frac{W^2}{Q_t} \left(\frac{3}{2C} \frac{Q_s}{M_0 UW} \left(\frac{1}{C} \frac{Q_s}{Q_t} - 1\right) - \frac{(\alpha-1)}{C} + (\alpha-2)\right)^{-1}$$

(40)

Here, $M_0$ is the minimum mass necessary to cover the bed per unit area, and $\alpha \approx 0.6$ is a dimensionless exponent that appears in the flow velocity equation (see eq. D4; Nitsche et al., 2012). The minimum mass $M_0$ can be evaluated by assuming that a single layer of closed-packed spherical grains resides on the bed (Turowski, 2009; Turowski and Hodge, 2017)

$$M_0 = \frac{\pi \rho_s D}{3\sqrt{3}}$$

(41)

### 2.3.3 Response time of bed cover

The response time for the adjustment of bed cover $T_C$ was previously derived by Turowski and Hodge (2017) and is given by

$$T_C = \frac{LM_0 W}{Q_t C}$$

(42)

### 2.3.4 Response time ratios

The dynamics of the channel during adjustment is to some extent determined by the relative magnitude of the response times. For example, if the response time for the adjustment of bed slope is always much longer than the response time for bed cover, on the time scale of slope adjustment, it can be assumed that bed cover is always at a steady state. The ratio of the response time for slope and width (widening channel) is given by

$$\frac{T_S}{T_W} = \frac{115\kappa}{9n} \frac{SL}{WC} \frac{Q_t}{Q_s} \frac{\theta}{\theta_c} \left(\frac{(\alpha-1)}{C} - \frac{3}{2C} \frac{Q_s}{M_0 UW} \left(\frac{1}{C} \frac{Q_s}{Q_t} - 1\right) - (\alpha-2)\right)$$

(43)

Similarly, for a narrowing channel

$$\frac{T_S}{T_N} = \frac{k_V ILSW}{nkQ_s C} (gS)^{\frac{1-\alpha}{2}} R^{\frac{1-3\alpha}{2}} \left(\frac{Q}{W}\right)^{\alpha-1}$$

(44)

The ratio of the response time for cover and slope is given by

$$\frac{T_C}{T_S} = \frac{gY}{230 k_v \sigma_T{}^2}\left(\frac{\rho_s}{\rho} - 1\right)\frac{n M_0}{S}\frac{Q_s}{Q_t}$$

(45)

The ratio of the response time for cover and width is given by

$$\frac{T_C}{T_W} = \frac{\kappa Y\left(\frac{\rho_s}{\rho} - 1\right)g M_0}{18 k_v \sigma_T^2}\frac{L}{WC}\frac{\theta}{\theta_c}\left(\frac{(\alpha - 1)}{C} - \frac{3}{2C}\frac{Q_s}{M_0 UW}\left(\frac{1}{C}\frac{Q_s}{Q_t} - 1\right) - (\alpha - 2)\right)$$

(46)

Similarly, for a narrowing channel

$$\frac{T_S}{T_N} = \frac{k_V I L M_0 W}{Q_t C}(gS)^{\frac{1-\alpha}{2}}R^{\frac{1-3\alpha}{2}}\left(\frac{Q}{W}\right)^{\alpha - 1}$$

(47)

## 3 Results

To illustrate the dependence of channel morphology and of the adjustment time scales on control and channel morphology parameters, I used parameter values oriented on Lushui at the Liwu River, Taiwan (Table 1; see Turowski et al., 2007). The values of reach parameters were either measured in the field or estimated using literature data. The value for discharge is representative for bedload-carrying flows, using the partitioning method proposed by Sklar and Dietrich (2006). The value of the exponent and prefactor of the flow velocity equation (D4) was selected using data by Nitsche et al. (2012).

Table 1: Parameter values used for the example calculations, following Turowski et al.'s (2007) estimates for the Liwu River, at Lushui, Taiwan.

| Parameter | Symbol | Value |
|---|---|---|
| *Material properties* | | |
| Density of water (kg/m$^3$) | $\rho$ | 1000 |
| Density of sediment (kg/m$^3$) | $\rho_s$ | 2650 |
| Young's modulus (MPa) | $Y$ | $5\times10^4$ |
| Rock tensile strength (MPa) | $\sigma_T$ | 10 |
| Rock resistance coefficient | $k_v$ | $10^6$ |
| *Constants in the equations* | | |
| Acceleration due to gravity (m/s$^2$) | $g$ | 9.81 |
| Flow velocity exponent | $\alpha$ | 0.6 |
| Flow velocity coefficient | $k_V$ | 1 |
| Bedload discharge exponent | $m$ | 1 |
| Bedload slope exponent | $n$ | 2 |
| Bedload coefficient (kg/m$^3$) | $K_{bl}$ | 11000 |
| Critical Shields stress | $\theta_c$ | 0.045 |
| Bedload fraction available for lateral erosion | $\kappa_T$ | 0.01 |
| Bar aspect ratio | $\kappa_{bar}$ | 5 |
| *Channel reach parameters* | | |
| Reach length (km) | $L$ | 10 |
| Channel bed slope | $S$ | 0.02 |
| Channel width (m) | $W$ | 40 |
| Median grain size (m) | $D$ | 0.04 |
| Roughness length scale (m) | $R$ | 0.2 |
| Water discharge (m$^3$/s) | $Q$ | 60 |
| Sediment supply (kg/s) | $Q_s$ | 200 |

## 3.1 Steady state channel morphology

The sideward deflection length scale $d$ is an important parameter setting channel morphology in steady state, in particular the channel width, which depends on the square root of $d$ (Turowski, 2018).

$$W = \sqrt{\frac{kQ_s d}{I}} = \sqrt{\frac{1.17 D \sin(\gamma)\left(\frac{\theta}{\theta_c} - 1\right) kQ_s}{I}}$$

5  (48)

Here, $d$ is estimated using saltation hop length of bedload particles over bare bedrock (eq. 25). Saltation hop length is dependent on the Shields stress, and the new formulation can consequently alter steady state scaling of channel width and slope. Unfortunately, equation (48) cannot be solved analytically, since Shields stress $\theta$ is non-linearly dependent on channel width and slope (see eq. D6), and a numerical solution is necessary (Fig. 4). As in the model by Turowski (2018), channel

10  width is independent of discharge (Fig. 4A) and the observed scaling between width and discharge arises from a co-dependence of discharge and sediment supply (see Fig. 4B).

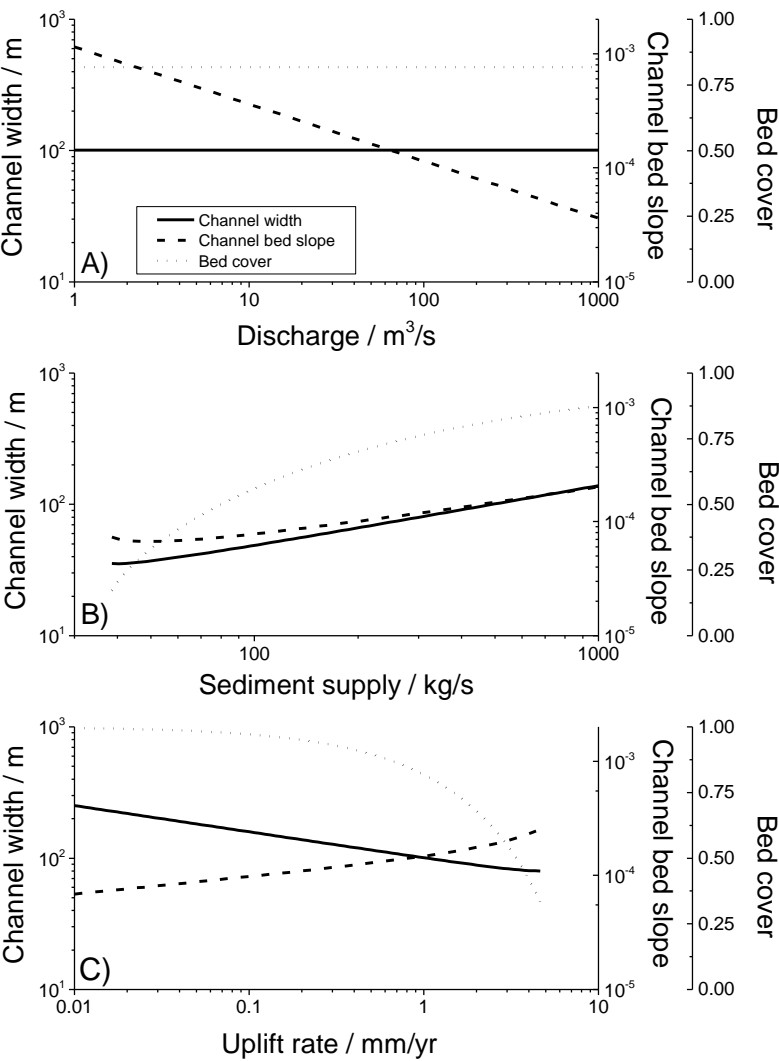

**Fig. 4: Steady state channel width (solid line), channel bed slope (dashed line), and bed cover (dotted line) against forcing variables discharge (A), sediment supply (B), and uplift rate (C). For the calculations, all other parameters have been kept constant (Table 1).**

### 3.2 Controls on adjustment timescales

For the calculation of adjustment timescales, the dependence of width and slope on discharge, sediment supply and uplift rate, and on each other, needs to be explicitly taken into account. From the derivation (App. C), the relevant width and slope in the time scale equations are those of the steady state morphology corresponding to the relevant control variables. As such, they are not independent of sediment supply, discharge, and other control variables. Within the model, steady state channel width and slope cannot be evaluated analytically, or written in a closed-form equation. Thus, a numerical solution is necessary. Adjustment time scales of width are generally longer than those for slope and for cover (Fig. 5), at least for the parameter values used in the example calculations (Table 1).

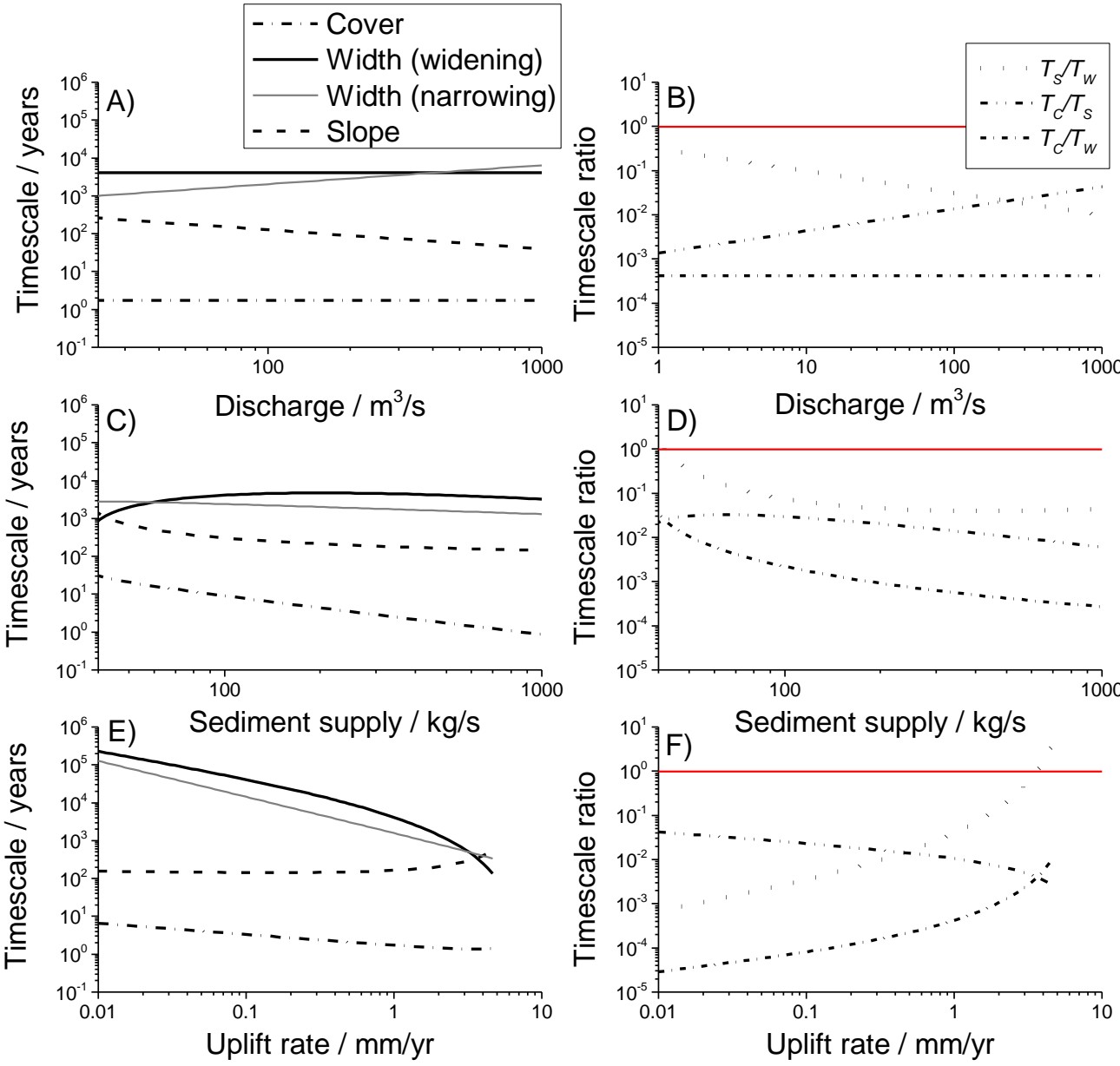

Fig. 5: Timescales (left column) and timescale ratios (right column) for channel adjustment, using appropriate steady state values corresponding to imposed discharge, sediment supply and uplift rate, for slope, width and cover, against forcing variables discharge (top row), sediment supply (middle row) and uplift rate (bottom row). For the calculations, all other parameters have been kept constant (Table 1). For the timescale ratios (B, D, F), only the timescale for widening channels was used, due to its similarity with the timescale for narrowing channels (A, C, E). The red solid line in the right column (B, D, F) indicates a ratio of one.

## 4 Discussion

### 4.1 Lateral erosion equation

Equation (24) is a mechanistic description of lateral fluvial bedrock erosion by impacting particles. Field and laboratory data that can be used to test the model are scarce, and the few data sets that exist do not include information on all necessary parameters to test it (e.g., Cook et al., 2014; Fuller et al., 2016; Suzuki, 1982; Mishra et al., 2018). The minimum parameters needed for a meaningful test are the lateral erosion rate measured in parallel with relevant driving variables including water discharge and bedload transport rate, in a channel with self-formed sediment cover and alternating gravel bars. Nevertheless, the model provides a starting point for future investigations, providing a clear mechanistic description and a host of testable assumptions and predictions.

Due to lack of direct relevant data and to keep the complexity of the model reasonable, it was necessary to make some assumptions on relevant processes and geomorphic response. For example, bedrock channels at high slopes tend to adjust their bed into a step-pool morphology (Duckson and Duckson, 1995; Scheingross et al., 2019). The feedbacks necessary to develop these bedforms, and how they may affect the flow hydraulics and erosion rates have not been considered in the present model (e.g., Scheingross and Lamb, 2017; Yager et al., 2012). In addition, it was necessary to quantify the wavelength of alternating bars. For the considerations on time scales presented here, the assumption of steady state cover had to be made, implying fully developed bars, and ignoring a potential braiding instability at large channel widths. Nelson and Seminara (2012) provided a linear stability analysis of bar formation over an initially bare bed. They stated explicitly that their considerations to not apply to the geometry of fully formed bars. However, their results and numerical model predictions by Inoue et al. (2016) could be interpreted to suggest that during the transient adjustment to fully formed bars from an initially empty bed under constant forcing conditions, bar wavelength varies little over time. Experimental evidence is rare. Some circumstantial observations can be found in the paper of Chatanantavet and Parker (2008), but these authors do not provide a systematic investigation or conclusive evidence for any type of scaling. In summary, none of the available studies was set up to investigate the controls of fully formed alternating bars, and a full understanding of the controls of their geometry is currently lacking. In absence of a full theory of alternating bars in bedrock channels, I have chosen to keep bar aspect ratio constant (eq. 21) in analogy with observations in alluvial channels (e.g., Kelly, 2006). Yet, due to the coupling with bed cover (eq. 23), this decision leads to unphysical behaviour in the limit of small degrees of cover. In this case, the bar wavelength is small, implying small bar width in comparison to channel width. As a consequence, the meandering bedload path has a large amplitude in comparison to its wavelength, and the deflection angle $\gamma$ approaches 90°. The assumption about bar wavelength is a minor piece in the model, affecting only the response time channel widening, which is linearly dependent on bar aspect ratio. For a full treatment of bar wavelength, we can speculate on the behaviour in two limits. First, at low values of cover, bar wavelength should be independent of cover, and is likely controlled by channel width or depth. Second, at high cover values, neighbouring bars start to overlap and the relationship to cover likely becomes more complicated. Further theoretical and experimental investigations are necessary to resolve this issue.

The lateral erosion equation (eq. 24) generally aligns with expected relations. Lateral erosion rates increase with increasing shear stress, sediment supply, and erodibility. However, they are inversely proportional to bed cover. This negative relationship arises because gravel bars increase their length as cover increases, due to their constant aspect ratio (eq. 23). This leads to less frequent impacts on the wall by travelling bedload. Fuller et al. (2016) observed that bedrock wall erosion is positively correlated with bed roughness in laboratory experiments. Similarly, Beer et al. (2017) observed higher wall erosion rates next roughness elements in a field study. The data from both of these papers are not sufficient for constraining a functional relationship between roughness and lateral erosion rates. In the model, lateral erosion rate (eq. 24) depends implicitly on

roughness, with a positive relationship, via the dependence on shear stress (see eq. D6). A similar implicit dependence can be found for the sideward deflection distance $d$ (eq. 25). Nevertheless, dedicated data on sideward deflection distances are needed to test the current equations and to guide future theoretical developments. Another aspect that is lacking in the current formulation is the dependence of lateral erosion rate on channel curvature. Recent work has attempted to address this within the stream-power framework of bedrock erosion (e.g., Langston and Tucker, 2018; Limaye and Lamb, 2014). Including channel curvature into the present model needs further work on bar deposition and bedload paths within curved channels (cf. Bunte et al., 2006; Fernandez et al., 2019; Mishra et al., 2018; Turowski, 2018).

## 4.2 Steady state channel morphology

In comparison to the model by Turowski (2018), the sideward deflection length scale $d$ has been explicitly quantified in terms of hydraulics (eq. 25), which may alter steady state relationships in comparison to the previously published model. In general, the updated model's predictions align with the results of Turowski (2018). It is somewhat surprising that channel width, like in Turowski's (2018) model, is explicitly independent of discharge (Fig. 4A), and, instead, is set by sediment supply (Fig. 4B). This implies that channel bed slope adjusts to changes in discharge without an effect on channel width, as long as sediment supply stays constant. The results arise because slope and discharge only feature in the same two equations, in that for Shields stress (eq. D6) and that for the bedload transport capacity (eq. C7). Using common parameter values for the relevant exponents $m$, $n$ and $\alpha$, the relationship between slope and discharge is the same in these two equations, allowing the two parameters to co-vary without affecting other parameters. Considering all other parameters constant, the first of these (eq. D6) gives the relation

$$S \sim Q^{\frac{2(\alpha-1)}{\alpha+1}}$$

(49)

while the second one (eq. C7) gives the relation

$$S \sim Q^{-\frac{m}{n}}$$

(50)

With the common parameter choice of $\alpha = 0.6$ (see Nitsche et al., 2012), $m = 1$, and $n = 2$ (see Turowski, 2018), we find that the two exponents are equal

$$\frac{2(\alpha - 1)}{\alpha + 1} = -\frac{m}{n} = -\frac{1}{2}$$

(51)

Thus, a change in discharge can be offset by a change in slope, without the need to vary any of the other parameters. Matthematically, this means that by substitution, the number of parameters and equations each can be reduced by one, and slope can be eliminated. A different choice of the $m/n$ ratio or of $\alpha$ would yield a direct dependence of width on discharge, and a dynamic co-evolution of slope and width.

## 4.3 Order principles and grade in bedrock channels

The condition of grade can be stated as what I call an order principle, which is a principle after which a dynamic system adjusts state variables to comply with forcing variables. Considering a stream without tributaries or hillslope sediment supply, the order principle for the condition of grade can be stated as follows: A river adjusts such that sediment flux is constant along the stream. The order principle is a direct consequence of the description of the sediment mass balance of the stream (see section 1).

Unlike alluvial channels, which feature a single type of material (the alluvium), in bedrock channels we need to also consider bedrock. This necessitates a second mass balance equation for bedrock (eq. 4), in addition to that for alluvium (see section 2.1). Accepting that a sediment mass balance cannot be neglected for a mechanistic description of bedrock channel dynamics, a bedrock river thus adjusts to two order principles, rather than one. The first of these is related to the mass balance of sediment (section 2.1) and leads to a condition of grade, as discussed above. The second of these is related to the mass balance of bedrock (eq. 4) and can be stated as follows: The river adjusts such that the vertical erosion rate is equal to the uplift or baselevel lowering rate.

When control variables change, the river responds by adjusting its morphology – slope, width, and bed cover – to comply with both of the order principles. However, due to the different adjustment time scales, the path to a new steady state morphology may be complex. As an example, consider a river at steady state, when sediment supply increases. The river responds by depositing sediment, increasing stationary sediment mass (eqs. 6 and 7). The increase in available stationary sediment increases entrainment rates (cf. Turowski and Hodge, 2017). Deposition continues until the river reaches a graded state in which sediment outflux from the considered reach is equal to sediment supply (eqs. 6 and 7). At the same time, any change in stationary sediment directly affects bed cover (eq. 32), and the immediate response of the stationary sediment mass is reflected in the short response times of bed cover (eq. 42; Fig. 5). Changes in cover, in turn, affect both vertical and lateral incision rates, initiating slope and width adjustment. These adjust much more slowly than bed cover (Fig. 5) until the vertical erosion rate matches the uplift rate. Yet, adjustments in width and slope feed back into the sediment dynamics, for example by affecting transport capacity. Again, the river responds by depositing or entraining material to maintain grade. The mutual feedback continues until both order principles – grade and the erosional balance with matching incision and uplift rates – are satisfied.

With two order principles controlling bedrock channel adjustment, the river may be in a steady state with respect to one of them but not with the other. Because the adjustment time scale for cover is shortest (Fig. 5), with values that range from minutes to days, it can be expected that bedrock rivers are close to a graded state most of the times (cf. Phillips and Jerolmack, 2016). Given the long adjustment times for width and slope, this does not necessarily mean, however, that they are in a steady state with respect to bedrock elevation, where incision rate matches uplift rate.

## 4.4 What is a bedrock channel?

The considerations and arguments presented in this paper affect the conceptual view of a bedrock channel, and the use of relevant terminology. We can distinguish detachment-limited and transport-limited channels, which are identified with the two end member descriptions focusing on the mass balance description of bedrock (detachment-limited) and sediment (transport-limited), respectively (cf. Shobe et al., 2017). For detachment-limited channels, we assume that the transport of sediment (eq. 6) can be neglected, i.e., sediment transport does not significantly impact channel dynamics and morphology. Formally, this assumption is valid if sediment supply is very much smaller than transport capacity, or stationary sediment mass $M_s \sim 0$. For transport-limited channels we assume that bedrock incision can be neglected (eq. 4). Formally, this assumption is valid if deposition or erosion has a negligible effect on the stationary sediment mass, in the mathematical limit as $M_s$ goes to infinity. The latter point implies that entrainment or deposition of sediment does not significantly affect stationary sediment mass.

A formal definition of bedrock channels should fulfil a number of criteria (cf. Turowski et al., 2008b). First, the definition should comply with the intuition of field workers. Alluvial and bedrock channels are end members on a continuum of channel types, and therefore, there will always be debated cases. But generally, most geomorphologists would agree whether the particular river is classified as an alluvial or bedrock river when seeing it in the field. Second, it should not rely on observations

of field parameters that can change quickly, for example over a single flood. Third, a useful definition should not rely on parameters that cannot be measured. Fourth, it should not rely on theoretical concepts that are untested, untestable, or debated. Fifth, a definition rooted in the understanding of relevant processes or dynamics is preferable to one that relies solely on descriptions of morphology.

Bedrock channels, in general, have often been classified as detachment-limited channels, in which long-term sediment-supply is (much) smaller than long-term sediment transport capacity (e.g., Whipple, 2004; Whipple et al., 2013). Further, this condition is generally assumed to result in partial sediment cover and exposed bedrock on channel bed and banks. Bedrock exposure in the channel can easily be observed in the field, and is therefore often used for channel classification (e.g.,
Montgomery et al., 1996; Tinkler and Wohl, 1998). A number of formal definitions of bedrock channels have been put forward based on these considerations. Exemplary, I will quote and discuss the most recent definition of Whipple et al. (2013):

> *Bedrock rivers may satisfy either or both of the following conditions: (1) the long-term capacity of the river to transport bedload (Qc) exceeds the long-term supply of bedload (Qs), resulting in generally sediment-starved conditions, significant rock exposure in bed and banks, and only thin, patchy, and temporary alluvial*
> *cover; or (2) the river is, over the long term (millennial to geologic timescales), actively incising through in-place rock.*

Few geomorphologists would argue against the second part of the definition, although it may difficult to assess this aspect in the field. Nevertheless, it is the first part of the definition that is relevant to the points made here, and which I reject based on the following general arguments and on the concepts developed in the present paper. First, the definition is theoretically laden
in the sense that a theoretical concept is imposed and equated to a field observation. To my knowledge, no methods currently exist that allow to reliably measure either long-term sediment supply or transport capacity. Even the inaccurate estimates that are currently possible need extensive field and modelling work, partly require strong assumptions, and are subject to large errors (e.g., Schneider et al., 2015). As such, the statement is not useful for the identification of bedrock channels in the field. Second, using mass balance arguments, I have demonstrated that bedrock channels adjust to a graded state. Unlike alluvial
rivers, this does not imply that sediment supply is equal to transport capacity. Rather, the relationship between cover and the ratio of supply and capacity is modulated by the deposition and entrainment of stationary sediment mass, i.e., bed cover. The model of Turowski and Hodge (2017) predicts partial cover for sediment supply values larger than transport capacity in some parameter configurations. Similarly, the simulations of Inoue et al. (2016) predict partially covered bed for conditions where sediment supply equals transport capacity. This shows that, depending on the theoretical formulation and the relevant concepts,
assumptions, and definitions, sediment supply values equal to or larger than transport capacity may be possible for bedrock channels. Third, even if the long-term sediment supply is lower than transport capacity, alluvial cover is not necessarily thin, patchy or temporary, as is assumed in the definition. Rather, there can be thick, substantial, widespread or persistent cover in the channel. For example, Shepherd (1972) and Fernandez et al. (2019) documented persistent gravel bars in experimental meandering bedrock channels. Theoretical cover models (e.g., Hodge and Hoey, 2012; Sklar and Dietrich, 2004; Turowski
and Hodge, 2017) predict substantial cover for certain sediment supply values that are smaller than the transport capacity. Experimental observations of run-away alluviation (e.g., Chatanantavet and Parker, 2008) provide evidence for this. Fourth, in a natural channel, sediment supply and discharge vary over timescales that are short in comparison to the adjustment timescales of channel width and slope. Upscaling discharge variability and sediment supply with a numerical model, Lague (2010) showed that the channel bed is either fully covered or sediment-free for the majority of the time. Long-term mean cover
values in his simulations exceeded a value of 0.5 in all cases, prohibiting the use of a detachment- or transport-limited approximation. Taken together, the arguments suggest that the connection between patchy, thin, and temporary alluvial cover and a ratio of sediment supply to transport capacity smaller than one is not tenable. As such, the definition, as proposed, is neither useful nor does it reflect current knowledge of bedrock channel dynamics. Turowski et al. (2008b) proposed an

alternative definition, stating that *a bedrock channel cannot substantially widen, lower, or shift its bed without eroding bedrock*. This definition has been discussed and slightly altered by Meshkova et al. (2012). It does not stand in contradiction to field observations, current process knowledge and newly emerged concepts, and can be readily applied in the field (see Turowski et al. 2008b for relevant field criteria).

## 5. Conclusions

Bedrock channel dynamics are controlled by two dominant order principles. They adjust their morphology both to achieve grade, in which the sediment transport rate is constant along the stream, and to match incision rate to uplift or baselevel lowering. The recognition of a steady state corresponding to one of these principles does not necessarily imply that the other has also been achieved. With minutes to days, the adjustment timescale for bed cover is short relative to the timescales for channel width and slope, and cover may be adjusted to changing supply conditions even over the duration of a single flood event. Thus, it can be expected that bedrock channels are close to a graded state most of the time. In the example calculations (Fig. 5, 6), adjustment timescales for slope and width are of the order of thousands of years. This is shorter than the major cyclic variations of Earth's climate (e.g., Roe, 2006), or the typical timescales of mountain building. The results therefore suggest that many bedrock channels are also close to an erosional steady state, in which erosion rate is equal to uplift rate.

## Appendix A: Deriving the Exner equation from the entrainment-deposition framework

Substituting eq. (7) into eq. (6) to eliminate entrainment and deposition rates, we obtain

$$\frac{\partial M_m}{\partial t} = -\frac{\partial q_s}{\partial x} - \frac{\partial M_s}{\partial t}$$

(A1)

Rearrange to get

$$\frac{\partial (M_m + M_s)}{\partial t} = -\frac{\partial q_s}{\partial x}$$

(A2)

Define a total sediment mass per unit area $M_{tot} = M_m + M_s$ and divide by the sediment density $\rho_r(1\text{-}p)$ to obtain the Exner equation

$$\frac{1}{\rho_r(1-p)}\frac{\partial M_{tot}}{\partial t} = \frac{\partial h_s}{\partial t} = -\frac{1}{\rho_r(1-p)}\frac{\partial q_s}{\partial x}$$

(A3)

## Appendix B: Estimating the deflection angle

Assume that the bedload particle path through the channel follows a sinusoidal path with a wavelength equal to the gravel bar spacing and an amplitude $A_{bar}$

$$y = A_{bar}\sin\left(2\pi\frac{x}{\lambda}\right)$$

(B1)

Here, $y$ denotes the distance in the cross-channel direction, with the channel centre line located at $y = 0$, and $x$ denotes the distance in the long-channel direction. The tangent of the angle $\gamma$ is given by the derivative of B1

$$\tan(\gamma) = \frac{dy}{dx} = 2\pi\frac{A_{bar}}{\lambda}\cos\left(2\pi\frac{x}{\lambda}\right)$$

(B2)

We are interested in the deflection angle $\gamma$ at the edge of the gravel bar, a distance $W_{covered}$, the covered part of the channel width, from the channel boundary, which corresponds to $y = W/2 - W_{covered}$. Hence, at the corresponding $x$-position $x_{edge}$

$$2\pi\frac{x_{edge}}{\lambda} = \sin^{-1}\left(\frac{W/2 - W_{covered}}{A_{bar}}\right)$$

(B3)

Here, $\sin^{-1}$ denotes the inverse sinus function. Combining equations B1-B3, and writing the path amplitude as a fraction $f = 2A/W$ of the half channel width, we obtain

$$\sin(\gamma) = \sin\left\{\tan^{-1}\left[\pi\frac{fW}{\lambda}\cos\left(\sin^{-1}\left(\frac{1}{f} - \frac{2W_{covered}}{fW}\right)\right)\right]\right\}$$

(B4)

Here, $\tan^{-1}$ denotes the inverse tangent function. Substituting eq. (23) for $\lambda$ and $C$ for $W_{covered}/W$ (eq. 22), we obtain

$$\sin(\gamma) = \sin\left\{\tan^{-1}\left[\frac{f\pi}{k_{bar}C}\cos\left(\sin^{-1}\left(\frac{1}{f} - \frac{2C}{f}\right)\right)\right]\right\}$$

(B5)

Assuming $f = 1$, a reasonable approximation for the square of B5 (as it appears in all equations) is

$$\sin^2(\gamma) \approx 1 - C$$

(B6)

**Appendix C: Deriving the response time scale of width adjustment using perturbation analysis**

For the following analysis we assume that all parameters are kept constant apart from sediment supply, which varies sinusoidally over time. This choice allows to obtain an analytical solution for the problem, and does not affect the result for the timescale of transient adjustment. Sediment supply can then be written as the sum of the average supply $\overline{Q_s}$ and a perturbation term $\delta Q_s$. The variation of the latter is described with a sinusoidal oscillation around zero.

$$Q_s = \overline{Q_s} + \delta Q_s$$

(C1)

$$\delta Q_s = K \sin\left(\frac{2\pi t}{P}\right)$$

(C2)

Here, $K$ is a constant and $P$ the period of the perturbation. Using linearized approximations to the differential equations (i.e., using first-order Taylor series to approximate non-linear functions), we then derive the width response to this perturbation, which can also be written as the sum of a time-independent term $\overline{W}$ and a time-dependent term $\delta W$.

$$W = \overline{W} + \delta W$$

(C3)

To obtain an equation describing the time evolution of channel width, we combine equations (24) and (27) to obtain:

$$\frac{dW}{dt} = \frac{2\kappa Y g}{k_v \sigma_T^2}\left(\frac{\rho_s}{\rho} - 1\right)\sin^2(\gamma)\frac{Q_s}{WC}\frac{\left(\frac{\theta}{\theta_c} - 1\right)}{\left(\frac{\theta}{\theta_c} - 1\right) + 24}$$

(C4)

We substitute the squared sine of the angle by (B6) to obtain

$$\frac{dW}{dt} = \frac{2\kappa Y g}{k_v \sigma_T^2}\left(\frac{\rho_s}{\rho} - 1\right)\frac{1 - C}{C}\frac{Q_s}{W}\frac{\left(\frac{\theta}{\theta_c} - 1\right)}{\left(\frac{\theta}{\theta_c} - 1\right) + 24}$$

(C4)

To simplify the equation further, I make the assumption that excess transport stage $\theta/\theta_c$ rarely exceeds the value of ten. Then, we can approximate (cf. Auel et al., 2017a):

$$\frac{\left(\frac{\theta}{\theta_c} - 1\right)}{\left(\frac{\theta}{\theta_c} - 1\right) + 24} \approx \frac{1}{36}\frac{\theta}{\theta_c}$$

(C5)

The width evolution equation is then

$$\frac{dW}{dt} = \frac{\kappa Y g}{18 k_v \sigma_T^2}\left(\frac{\rho_s}{\rho} - 1\right)\frac{1 - C}{C}\frac{Q_s}{W}\frac{\theta}{\theta_c}$$

(C6)

Steady state cover can be described with equation (32) (Turowski and Hodge, 2017).

$$C = \left(1 - e^{-\frac{Q_s}{M_0 U W}}\right)\frac{Q_s}{Q_t}$$

(C6)

The bedload transport capacity can be written using eq. (33) (see Turowski, 2018)

$$\frac{Q_t}{W} = K_{bl} Q^m S^n$$

(C7)

$$\frac{dW}{dt} = AK_{bl}Q^m S^n \left(1 - e^{-\frac{Q_s}{M_0 UW}}\right)^{-1} (W)^{\alpha-1} - AQ_s(W)^{\alpha-2}$$

(C8)

With

$$A = \frac{\kappa Y g}{18 k_v \sigma_T^2} \left(\frac{\rho_s}{\rho} - 1\right) \frac{(gS)^{\frac{\alpha+1}{2}} R^{\frac{3\alpha-1}{2}}}{k_V \left(\frac{\rho_s}{\rho} - 1\right) D\theta_c} (Q)^{1-\alpha}$$

5  (C9)

Here, to reduce the number of parameters and reveal implicit dependencies, the Shields stress has been substituted using standard hydraulic scaling relations (Appendix D). The parameter $\alpha$ is a dimensionless constant that typically takes a value of 0.6, and is a measure of roughness with the dimensions of length (see Nitsche et al., 2012). Next, eqs. C1 and C3 are substituted into C4, and expanded using first-order Taylor approximations of the form

$$(B(\bar{W} + \delta W)^{\alpha-1} - 1) \approx (B\bar{W}^{\alpha-1} - 1) + (\alpha - 1)B\bar{W}^{\alpha-2}\delta W$$

 (C10)

$$\left(1 - e^{-\frac{Q_s}{M_0 UW}}\right)^{-1} \approx \left(1 - e^{-\frac{\overline{Q_s}}{M_0 U\bar{W}}}\right)^{-1} - \left(1 - e^{-\frac{\overline{Q_s}}{M_0 U\bar{W}}}\right)^{-2} \left[\frac{1}{M_0 U\bar{W}} e^{-\frac{\overline{Q_s}}{M_0 U\bar{W}}} \delta Q_s + \frac{\overline{Q_s}}{M_0} \left(\frac{1}{U^2\bar{W}}\frac{\partial U}{\partial W} + \frac{1}{U\bar{W}^2}\right) e^{-\frac{\overline{Q_s}}{M_0 U\bar{W}}} \delta W\right]$$

(C11)

After some algebra and dropping terms that are quadratic or cubic in the delta terms $\delta Q_s$ and $\delta W$, we obtain

$$\frac{d\delta W}{dt} = AK_{bl} \left(1 - e^{-\frac{\overline{Q_s}}{M_0 U\bar{W}}}\right)^{-1} (\bar{W})^{\alpha-1} - A\overline{Q_s}(\bar{W})^{\alpha-2}$$

$$+ \left[AK_{bl}Q^m S^n \left(1 - e^{-\frac{\overline{Q_s}}{M_0 U\bar{W}}}\right)^{-1} (\alpha-1)\bar{W}^{\alpha-2}\right.$$

$$\left. - AK_{bl}Q^m S^n \left(1 - e^{-\frac{\overline{Q_s}}{M_0 U\bar{W}}}\right)^{-2} \frac{\overline{Q_s}}{M_0} \left(\frac{1}{U^2\bar{W}}\frac{\partial U}{\partial W} + \frac{1}{U\bar{W}^2}\right) e^{-\frac{\overline{Q_s}}{M_0 U\bar{W}}} \bar{W}^{\alpha-1} - A(\alpha-2)\overline{Q_s}\bar{W}^{\alpha-3}\right]\delta W$$

$$+ \left[AK_{bl}Q^m S^n \left(1 - e^{-\frac{\overline{Q_s}}{M_0 U\bar{W}}}\right)^{-2} \frac{1}{M_0 U} e^{-\frac{\overline{Q_s}}{M_0 U\bar{W}}} \bar{W}^{\alpha-2} - A\bar{W}^{\alpha-2}\right]\delta Q_s$$

(C12)

20  Resubstituting for cover, particle speed and so on, we obtain

$$\frac{d\delta W}{dt} = AK_{bl}\frac{\overline{Q_s}}{\bar{C}Q_t}(\bar{W})^{\alpha-1} - A\overline{Q_s}(\bar{W})^{\alpha-2} + A\overline{Q_s}\bar{W}^{\alpha-3}\left[\frac{(\alpha-1)}{\bar{C}} - \frac{3}{2\bar{C}}\frac{\overline{Q_s}}{M_0 U\bar{W}}\left(\frac{1}{\bar{C}}\frac{\overline{Q_s}}{Q_t} - 1\right) - (\alpha-2)\right]\delta W$$

$$+ A\bar{W}^{\alpha-2}\left[\frac{\overline{Q_s}}{M_0 U\bar{W}}\left(\frac{1}{\bar{C}}\frac{\overline{Q_s}}{Q_t} - 1\right) - 1\right]\delta Q_s$$

(C13)

Next, equation C2 is substituted in C13 to obtain a differential equation of the form

$$\frac{d\delta W}{dt} = K_1\left[K_2 + K\sin\left\{\frac{2\pi t}{P}\right\} + K_3\delta W\right]$$

(C14)

The general solution to C14 is

$$\delta W = \frac{K_1 K_2 \left(\frac{P}{2\pi}\right)}{K_1^2 K_3^2 \left(\frac{P}{2\pi}\right)^2 + 1} \sqrt{\left(K_1^2 K_3^2 \left(\frac{P}{2\pi}\right)^2 + 1\right)} \sin\left\{\frac{2\pi t}{P} + \varphi\right\} + \frac{KK_1^2 K_3^2 \left(\frac{P}{2\pi}\right)^2 + K}{K_1^2 K_3^2 \left(\frac{P}{2\pi}\right)^2 + 1} + c_1 \exp\{K_1 K_3 t\}$$

(C15)

Here, $c_1$ is the integrative constant and $\varphi$ is a phase shift of the width response to the perturbation in sediment supply. The exponential term describes transient adjustment to the steady state and can be used to obtain the response time.

$$T_W = -\frac{1}{K_1 K_3}$$

(C16)

Collecting the terms, we obtain

$$T_W = \frac{18 k_v \sigma_T^2}{\kappa Y g} \frac{k_V g D \theta_c}{(gS)^{\frac{\alpha+1}{2}} R^{\frac{3\alpha-1}{2}}} \frac{Q^{\alpha-1}}{\overline{Q_s}} \overline{W}^{3-\alpha} \left( \frac{3}{2\overline{C}} \frac{\overline{Q_s}}{M_0 U \overline{W}} \left( \frac{1}{\overline{C}} \frac{\overline{Q_s}}{Q_t} - 1 \right) - \frac{(\alpha-1)}{\overline{C}} + (\alpha-2) \right)^{-1}$$

(C17)

Equation C17 is considerably simpler in terms of shear stress

$$T_W = \frac{18 k_v \sigma_T^2}{\kappa Y \left( \frac{\rho_s}{\rho} - 1 \right) g} \frac{\theta_c}{\theta} \frac{\overline{W}^2}{Q_t} \left( \frac{3}{2\overline{C}} \frac{\overline{Q_s}}{M_0 U \overline{W}} \left( \frac{1}{\overline{C}} \frac{\overline{Q_s}}{Q_t} - 1 \right) - \frac{(\alpha-1)}{\overline{C}} + (\alpha-2) \right)^{-1}$$

(C18)

In the linear cover approximation (cover-dominated limit; see Turowski, 2018), we have

$$\overline{C} = \frac{\overline{Q_s}}{Q_t}$$

(C19)

Thus, (C18) becomes

$$T_{W,cover} = \frac{18 k_v \sigma_T^2}{\kappa Y \left( \frac{\rho_s}{\rho} - 1 \right) g} \frac{\theta_c}{\theta} \frac{\overline{W}^2}{Q_t} \left( \frac{1}{(\alpha-2) \frac{\overline{Q_s}}{Q_t} - (\alpha-1)} \right)$$

(C20)

**Appendix D: Writing shear stress and bedload speed in terms of discharge**

The development here is equivalent to that given by Turowski (2018).

The reach-averaged Shields stress $\theta$ is defined by

$$\theta = \frac{\tau}{(\rho_s - \rho) g D}$$

(D1)

Here, $\tau$ is the shear stress, given by the DuBoys equation

$$\tau = \rho g H S$$

(D2)

The continuity equation for water flow is

$$Q = WHV$$

(D3)

There a number of different equations available to compute water flow velocity $V$. For mountain streams, a discharge-based variable power flow resistance equation has been found to be a good description of available data (Ferguson, 2007; Nitsche et al., 2012)

$$V = k_V (gS)^{\frac{1-\alpha}{2}} R^{\frac{1-3\alpha}{2}} \left(\frac{Q}{W}\right)^{\alpha}$$

(D4)

Here, $R$ is a measure of bed roughness with dimensions of length, for example the standard deviation of the bed surface (e.g., Nitsche et al., 2012), and $k_V \approx 1$ and $\alpha \approx 0.6$ are constants. Combining C2, C3, and C4, shear stress

5   can be written as

$$\tau = \frac{\rho}{k_V} (gS)^{\frac{\alpha+1}{2}} R^{\frac{3\alpha-1}{2}} \left(\frac{Q}{W}\right)^{1-\alpha}$$

(D5)

The Shields stress is thus given by

$$\theta = \frac{(gS)^{\frac{\alpha+1}{2}} R^{\frac{3\alpha-1}{2}}}{k_V \left(\frac{\rho_s}{\rho} - 1\right) gD} \left(\frac{Q}{W}\right)^{1-\alpha}$$

10  (D6)

The downstream bedload velocity arises in the cover relation (eq. 32), and can be written as:

$$U = 1.46 \left( \left(\frac{\rho_s}{\rho} - 1\right) gD \right)^{1/2} \left(\frac{\theta}{\theta_c} - 1\right)^{1/2}$$

(D7)

In terms of discharge, this evaluates to

$$U = 1.46 \left( \left(\frac{\rho_s}{\rho} - 1\right) gD \right)^{1/2} \left( \frac{(gS)^{\frac{\alpha+1}{2}} R^{\frac{3\alpha-1}{2}}}{k_V \theta_c \left(\frac{\rho_s}{\rho} - 1\right) gD} \left(\frac{Q}{W}\right)^{1-\alpha} - 1 \right)^{1/2}$$

(D8)

**Notation**

| | | |
|---|---|---|
| | $A_{bar}$ | Bar amplitude [m]. |
| | $A_{cover}$ | Covered bed area [m²]. |
| | $A_{tot}$ | Total bed area [m²]. |
| 5 | $A_w$ | Actively eroding channel wall area [m²]. |
| | $a$ | Scaling exponent, $d$-$A$. |
| | $B$ | Constant in non-linear wave equation, describing slope development [m/s]. |
| | $b$ | Scaling exponent, $\beta$-$A$. |
| | $C$ | Fraction of covered bed. |
| 10 | $C_{SS}$ | Steady state cover. |
| | $c$ | Scaling exponent, $Q$-$A$. |
| | $c_s$ | Celerity of non-linear wave equation, describing slope development [m/s]. |
| | $d$ | Sideward deflection length scale, reach [m]. |
| | $D$ | Sediment deposition rate per bed area [kg/m²s]. |
| 15 | $D_{50}$ | Median grain size [m]. |
| | $e$ | Base of the natural logarithm. |
| | $E$ | Sediment entrainment rate per bed area [kg/m²s]. |
| | $f$ | Bedload path amplitude as fraction of channel width |
| | $F_{CD}$ | Cover-dependent term in the lateral erosion equation. |
| 20 | $F_T$ | Tools factor in the lateral erosion equation [kg/s]. |
| | $g$ | Acceleration due to gravity [m/s²]. |
| | $h_b$ | Bedrock elevation [m]. |
| | $h_s$ | Sediment elevation [m]. |
| | $H$ | Water depth [m]. |
| 25 | $I$ | Vertical erosion rate [m/s]. |
| | $K_{bl}$ | Bedload transport efficiency [kg m$^{-3m}$s$^{-m}$]. |
| | $k_e$ | Erodibility in stream power model [m$^{1-3m}$s$^{1-m}$]. |
| | $k_h$ | Hydrology coefficient [m$^{3-2c}$/s]. |
| | $k_{LE}$ | Lateral erosion coefficient [m/kg]. |
| 30 | $k_s$ | Steepness index [m$^{2\theta}$]. |
| | $k_{tools}$ | Lumped constant, tools-dominated channel slope. |
| | $k_v$ | Rock erodibility coefficient. |
| | $k_V$ | Velocity coefficient [m$^{2\alpha}$]. |
| | $L$ | Reach length [m]. |
| 35 | $M_0$ | Minimum mass per area necessary to cover the bed [kg/m²]. |
| | $M_m$ | Mobile sediment mass [kg/m²]. |
| | $M_s$ | Stationary sediment mass [kg/m²]. |
| | $m$ | Discharge exponent in bedload equation. |
| | $m'$ | Discharge exponent in the stream power model. |
| 40 | $n$ | Slope exponent in bedload equation. |
| | $n'$ | Slope exponent in the stream power model. |
| | $q_s$ | Mass sediment transport rate per unit width [kg/ms]. |
| | $q_t$ | Mass sediment transport capacity per unit width [kg/ms]. |

| | | |
|---|---|---|
| | $Q$ | Water discharge [m$^3$/s]. |
| | $Q_c$ | Critical discharge for the onset of bedload motion [m$^3$/s]. |
| | $Q_c^*$ | Relative sediment supply at the critical cover. |
| | $Q_s$ | Upstream sediment mass supply [kg/s]. |
| 5 | $Q_s^*$ | Relative sediment supply; sediment transport rate over transport capacity. |
| | $Q_t$ | Mass sediment transport capacity [kg/s]. |
| | $R$ | Bed roughness length scale [m]. |
| | $S$ | Channel bed slope. |
| | $S_{cover}$ | Channel bed slope predicted in the cover-dominated approximation. |
| 10 | $S_{tools}$ | Channel bed slope predicted in the tools-dominated approximation. |
| | $S_V$ | Valley slope. |
| | $T_C$ | Time scale of cover adjustment [s]. |
| | $T_N$ | Time scale of width adjustment, for a narrowing channel [s]. |
| | $T_S$ | Time scale of slope adjustment [s]. |
| 15 | $T_W$ | Time scale of width adjustment, for a widening channel [s]. |
| | $U$ | Bedload speed [m/s]. |
| | $V$ | Water flow velocity [m/s]. |
| | $W$ | Channel width [m]. |
| | $W_{cover}$ | Covered length within the channel width [m]. |
| 20 | $W_{uncover}$ | Uncovered length within the channel width [m]. |
| | $W_{ss}$ | Steady state channel width [m]. |
| | $x$ | Dimensional streamwise spatial coordinate [m]. |
| | $Y$ | Young's modulus of the bedrock [kg m$^{-1}$s$^{-2}$]. |
| | $\alpha$ | Scaling exponent, $V$-$Q$. |
| 25 | $\beta$ | Fraction of sediment transported as bedload. |
| | $\theta$ | Shields stress. |
| | $\theta_c$ | Critical Shields stress for the onset of sediment motion. |
| | $\lambda$ | Bar wavelength [m]. |
| | $\rho$ | Density of water [kg/m$^3$]. |
| 30 | $\rho_{bulk}$ | Bulk density of sediment [kg/m$^3$]. |
| | $P_r$ | Density of bedrock [kg/m$^3$]. |
| | $\rho_s$ | Density of sediment [kg/m$^3$]. |
| | $\sigma_T$ | Rock tensile strength [kg m$^{-1}$s$^{-2}$]. |
| | $\kappa$ | Lumped constant, width evolution equation [m$^{-2}$]. |
| 35 | $\kappa_{bar}$ | Bar aspect ratio. |
| | $\kappa_C$ | Coefficient in the cover term of width evolution. |
| | $\kappa_T$ | Coefficient in the tools term of width evolution. |
| | $\xi$ | Average sediment thickness above the bedrock [m]. |
| | $\tau$ | Bed shear stress [N/m$^2$]. |
| 40 | $\tau_c$ | Critical bed shear stress at the onset of bedload motion [N/m$^2$]. |

**Data availability**

No original data were used in this study.

**Competing interests**

The author declares that he has no conflict of interest.

**Acknowledgements**

I thank Claire Masteller, Aaron Bufe and Joel Scheingross for discussions. Ron Nativ provided detailed comments on an earlier version of the manuscript. Two anonymous reviewers provided constructive comments that helped to improve the paper.

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

Tables

**Table 1: Parameter values used for the example calculations, following Turowski et al.'s (2007) estimates for the Liwu River, at Lushui, Taiwan.**

| Parameter | Symbol | Value |
|---|---|---|
| *Material properties* | | |
| Density of water (kg/m$^3$) | $\rho$ | 1000 |
| Density of sediment (kg/m$^3$) | $\rho_s$ | 2650 |
| Young's modulus (MPa) | $Y$ | $5\times10^4$ |
| Rock tensile strength (MPa) | $\sigma_T$ | 10 |
| Rock resistance coefficient | $k_v$ | $10^6$ |
| *Constants in the equations* | | |
| Acceleration due to gravity (m/s$^2$) | $g$ | 9.81 |
| Flow velocity exponent | $\alpha$ | 0.6 |
| Flow velocity coefficient | $k_V$ | 1 |
| Bedload discharge exponent | $m$ | 1 |
| Bedload slope exponent | $n$ | 2 |
| Bedload coefficient (kg/m$^3$) | $K_{bl}$ | 11000 |
| Critical Shields stress | $\theta_c$ | 0.045 |
| Bedload fraction available for lateral erosion | $\kappa_T$ | 0.01 |
| Bar aspect ratio | $\kappa_{bar}$ | 5 |
| *Channel reach parameters* | | |
| Reach length (km) | $L$ | 10 |
| Channel bed slope | $S$ | 0.02 |
| Channel width (m) | $W$ | 60 |
| Median grain size (m) | $D$ | 0.04 |
| Roughness length scale (m) | $R$ | 0.2 |
| Water discharge (m$^3$/s) | $Q$ | 40 |
| Sediment supply (kg/s) | $Q_s$ | 200 |

Figures

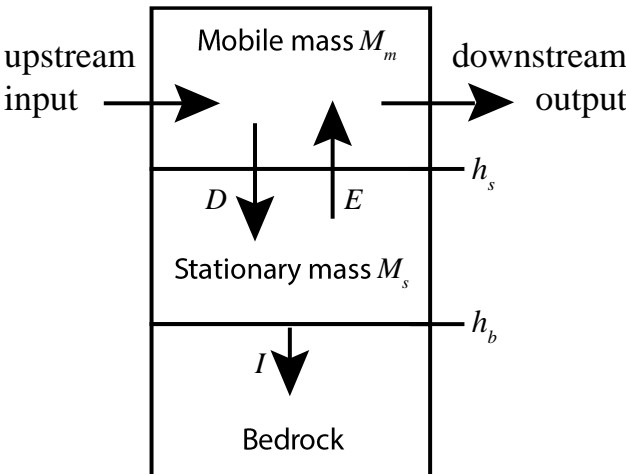

**Figure 1: Schematic side view of a control volume within a bedrock channel. The bedrock (bottom) is overlain by stationary sediment (centre), which exchanges particles via entrainment *E* and deposition *D* with the mobile sediment in the water column (top). The bedrock surface $h_b$ lowers at the incision rate *I*, while the sediment surface $h_s$ evolves according to the balance of entrainment and deposition (eq. 6).**

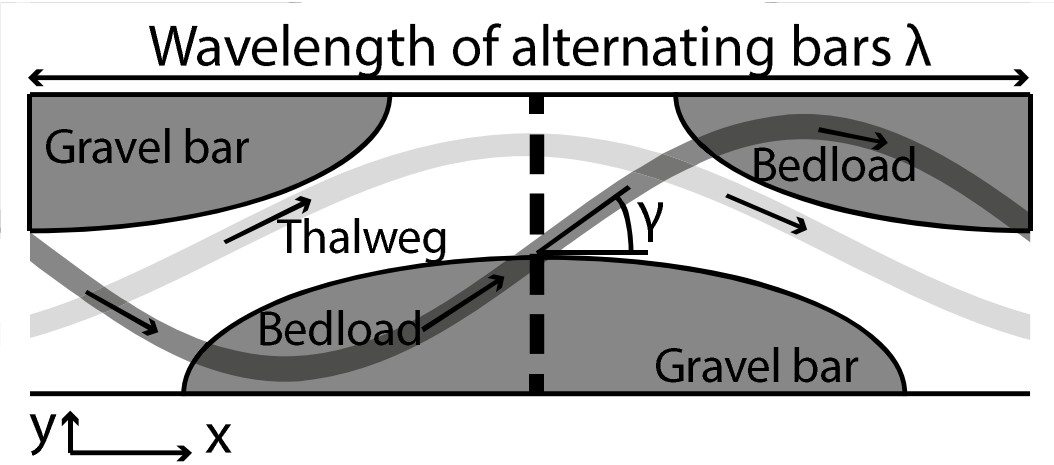

**Figure 2: Schematic top view of a straight bedrock channel, with alternating submerged gravel bars (dark grey) on a bedrock bed (white). The sinuous thalweg (light grey) and bedload path (transparent dark grey) are indicated. The black dashed line indicates the cross section that is ideal for sideward deflection of particles; here, the bedload particle stream crosses the boundary between gravel and smooth bedload. The wavelength of the alternating bars and therefore of the bedload path should scale with channel width. Adapted from Turowski (2018).**

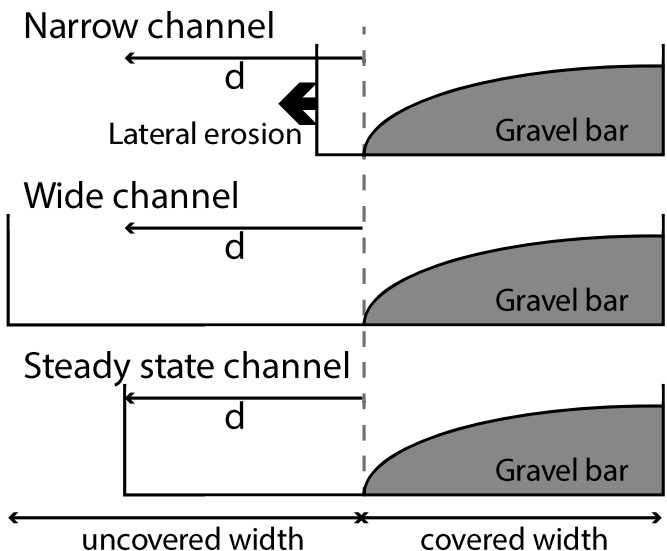

**Figure 3: The sideward deflection length scale *d* interacts with bed cover and channel width to determine whether the lateral erosion occurs (top), or not (center, bottom). Adapted from Turowski (2018).**

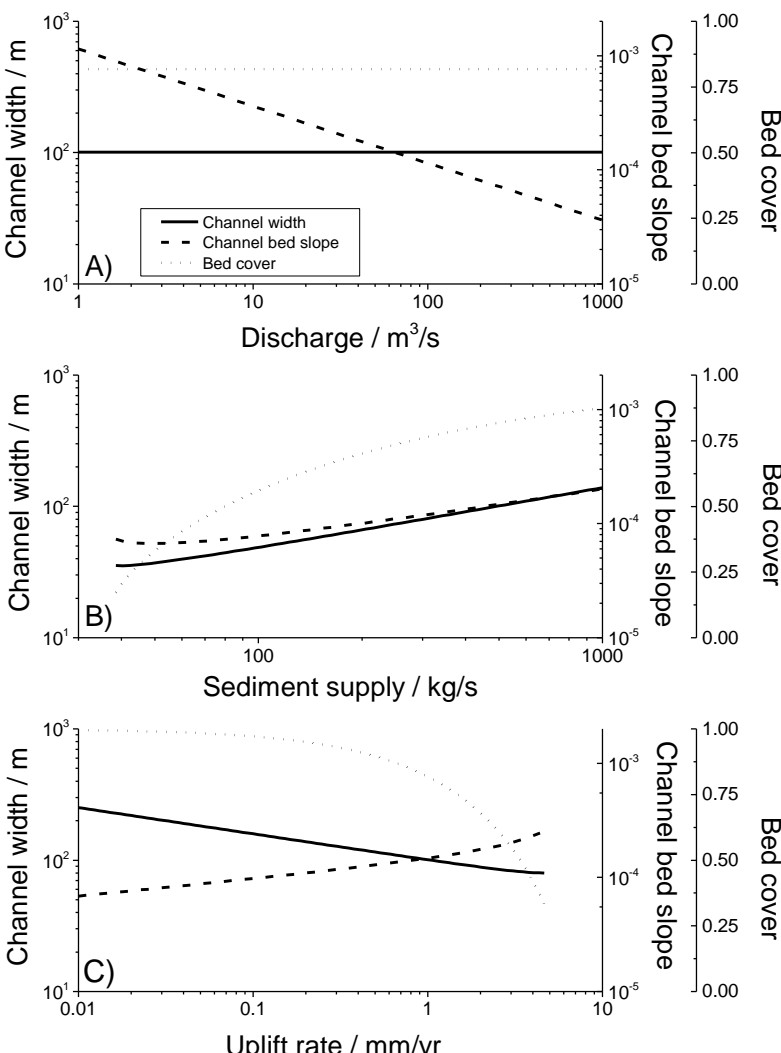

5    **Fig. 4: Steady state channel width (solid line), channel bed slope (dashed line), and bed cover (dotted line) against forcing variables discharge (A), sediment supply (B), and uplift rate (C). For the calculations, all other parameters have been kept constant (Table 1).**

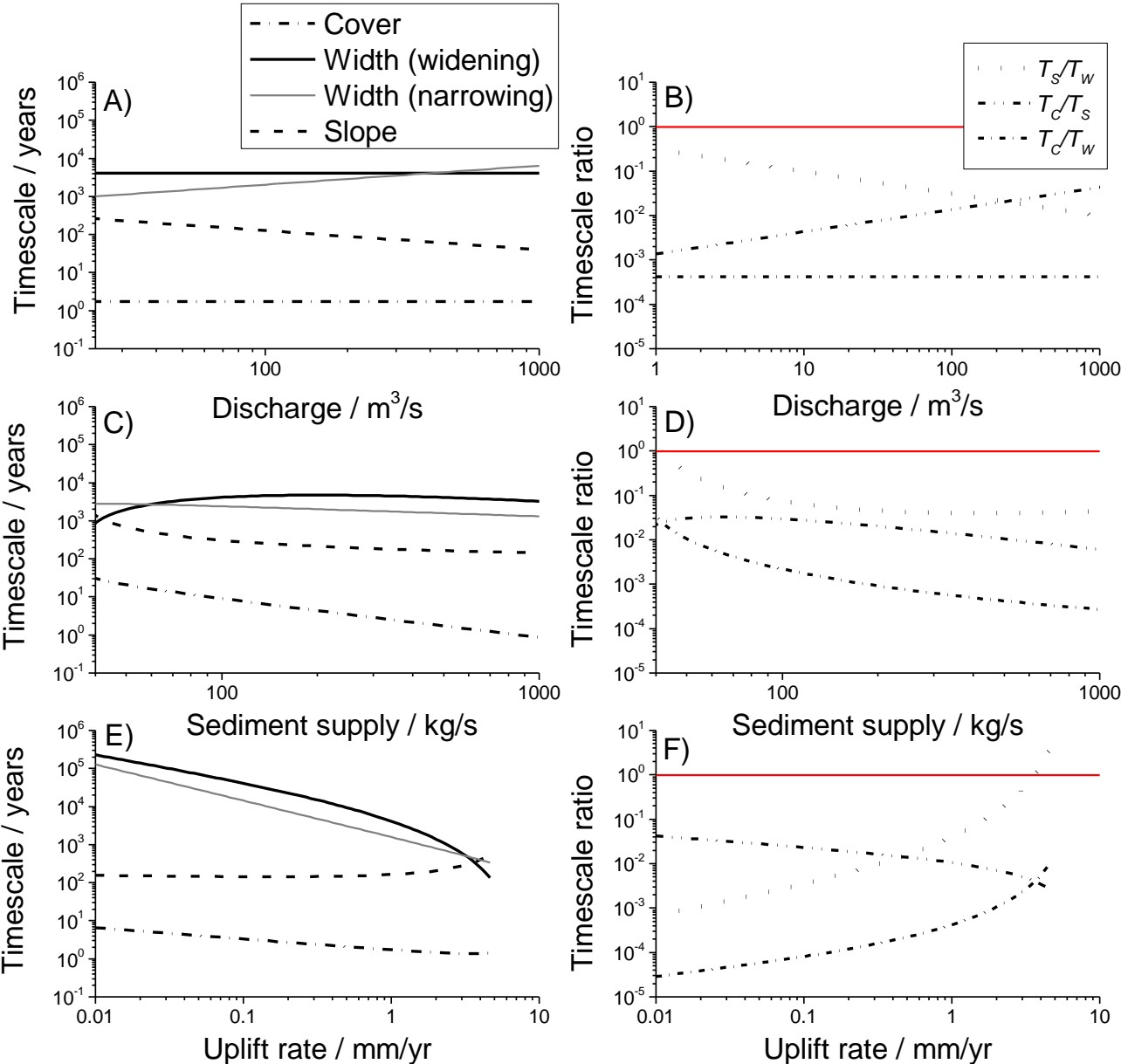

**Fig. 5: Timescales (left column) and timescale ratios (right column) for channel adjustment, using appropriate steady state values corresponding to imposed discharge, sediment supply and uplift rate, for slope, width and cover, against forcing variables discharge (top row), sediment supply (middle row) and uplift rate (bottom row). For the calculations, all other parameters have been kept**
5  **constant (Table 1). For the timescale ratios (B, D, F), only the timescale for widening channels was used, due to its similarity with the timescale for narrowing channels (A, C, E). The red solid line in the right column (B, D, F) indicates a ratio of one.**