# Peer review of "Mass balance, grade, and adjustment timescales in bedrock channels"

_Earth Surface Dynamics, 2019_

## Referee Comment (RC1) · Anonymous Referee #1 · 29 Oct 2019

In this manuscript, the author proposed a mechanistic model for analyzing the adjustment timescales for channel width, channel bed slope and alluvial bed cover in a mixed alluvial - bedrock channel. However, in the current version a significant question on the assumption of the bar wavelength remains and needs to be addressed.

In eq. 23, you assumed that the bar wavelength decreases with decreasing the fraction of alluvial cover. However, recent studies indicate that the bar wavelength increases with decreasing the fraction of alluvial cover in mixed alluvial – bedrock channel, in theoretically (Nelson and Seminara, 2011, Fig.2b, doi: 10.1029/2011GL050806) and numerically (Inoue et al., 2016, Figs 5 and 11, doi: 10.1061/(ASCE)HY.1943-7900.0001124.). Experiments conducted by Chatanantavet and Parker (2008) also show no decrease in bar wavelength. Your assumption is based on Kelly (2006)'s ob-

servations in alluvial channel, but may not be applicable to mixed alluvial – bedrock channel. Because this assumption directly affects the lateral erosion rate and the timescales, the results shown in Figs 4 – 6 may be incorrect.

Additional comments by line number below:

P2 Eq. (1): The density of the sediment?

P7 Line 9: Auel et al., 2017 a or b?

P7 Line 15: Why does the secondary flow not affect the lateral impact velocity?

P7 Line 32: Nelson and Seminara analyzed alternate bars in a mixed alluvial – bedrock channel in 2012, not 2011. The paper listed in the reference is probably incorrect.

P12 Figure 4c: There is no explanation of Fig4c in the text. Why does slope and width change with uplift rate?

P15. Line 6: Gravel bars do not increase their wavelength as cover increases.

P19 Eq. (B6): When C is close to 0 (i.e., almost completely exposed bedrock), $\gamma$ is close to 90 degree (i.e., sediments move towards the sidewalls). Why?

P26-P29: Inoue et al. (2014), Montgomery et al. (1996), Shepherd (1972) and Whipple (2004) are not listed in the references.

---

## Author Comment (AC1) · 30 Oct 2019

I thank the reviewer for taking to the time to read and comment on my paper. I am very grateful for the comments. This is a quick reply to the reviewer's comments on bar wavelength and its relationship to cover. I have looked again at the suggested papers and give a short summary below. Hopefully, this can instigate some further exchange before the discussion closes. In case I have overlooked anything, I am happy to receive further pointers.

The reviewer is unhappy with the relationship between cover and bar wavelength used in the model, stating that "Gravel bars do not increase their wavelength as cover increases" (comment to page 15 line 6, emphasized elsewhere). She/he cites three

papers as support, two modelling papers by Nelson and Seminara, 2011, and by Inoue et al., 2016, as well as an experimental study by Chatanantavet and Parker, 2008.

As a reply, first it needs to be noted that a change in the bar geometry affects only the lateral erosion equation (eq. 24 in the paper). This propagates into the response time of width for a widening channel, but not the response times of cover, of bed slope or for the width for a narrowing channel. It also does not affect the steady state channel morphology presented in Fig. 4. Without having done the necessary calculations, I expect that the effects on magnitude and scaling of response times will be minor. In summary, a change in the dependence of bar wavelength on cover would not majorly change the overall arguments and conclusions of the paper.

Second, looking through the suggested papers, I cannot find material supporting the reviewer's statement. I will briefly summarize the relevant points below.

The experiments of Chatanantavet and Parker (2008) were not set up to study bar morphology. They mention that alternating bars were present in the experiments (e.g., paragraph 13), but do not give details on their morphology or how their wavelength scales with cover. As such, the paper does not contain relevant information beyond the observation that alternating bars were present.

The Nelson and Seminara (2011) paper is not concerned with bars. However, there is another paper by the same authors, Nelson and Seminara (2012), which is. Consequently, I assume that the reviewer intended to refer to this particular study.

The two mentioned modelling studies (Nelson and Seminara, 2012, and Inoue et al., 2016) use slightly different modelling approaches, but reach similar conclusions. In fact, with regards to bar wavelength, Inoue et al. (2016) explicitly summarized the results of both as follows (page 8, left-hand column, 2nd full paragraph): "Nelson and Seminara (2012) conducted a linear stability analysis of bars on the bedrock and analyzed the wavelength of infinitesimal bars. The findings of their analysis are as follows: (1) regions where alternate bars form on the bedrock are determined not only by the

width/depth ratio, but to some degree by the ratio of $\tau = \tau$ c; and (2) the wavelength of the bars increases with decreasing sediment supply rate. The analysis by Nelson and Seminara (2012), unlike the simulation of this study, did not consider localized bedrock erosion by bedload; therefore, it is not possible to compare the two simulations quantitatively. However, the two models show a similar tendency to form longer wavelength bars when the sediment supply is lower." I draw attention to the concluding sentence of the paragraph, where Inoue et al. (2016) stated that bars feature longer wavelength for lower sediment supply. The latter condition implies less cover. As such, the statement is in contradiction to the claims of the reviewer, and in qualitative agreements with my assumption.

That said, the reviewer draws attention to an unphysical behavior that I had not previously noticed: Due to the linear relationship, at low cover, bar wavelength becomes short. This is in contradiction to the analysis of Nelson and Seminara (2012), who demonstrated that short bars are unstable and cannot persist (their Fig. 3). Furthermore, it leads to bedload paths at $90°$ angles to the main line of the channel when predicted bar wavelength is much smaller than channel width (pointed out by the reviewer in the comment to page 19, eq. B6). As a consequence, for low sediment supply rates, either the bar wavelength needs to be fixed in accordance with the stability criteria suggested by Nelson and Seminara (2012), or bars would disappear entirely, implying that lateral erosion is not possible within the assumptions of the model.

I will add a discussion of these points to the revised paper.

With best wishes and thanks for reviewing the paper, Jens Turowski

---

## Referee Comment (RC2) · Anonymous Referee #1 · 31 Oct 2019

In linear stability analysis, "unstable" means that the initial small bar height grows (i.e., bars can form). In Nelson and Seminara (2012), $\lambda$ is the wavenumber (=$2\pi$B/L, L is the wavelength). Therefore, Fig.3 shows that long bars are unstable and can form, or short bars are stable and can not form. Fig. 2 shows that growth rate of long bars increases with decreasing sediment supply. This imply that long bars are more likely to form when the alluvial cover is low. In your assumption (eq. 23), bar wavelength decreases with decreasing alluvial cover. You also stated that "gravel bars increase their length as cover increases" in page 15 line 6. These are in contradiction to Nelson and Sminara (2012).

In numerical analysis (Inoue et al., 2016), the thickness of gravel bars decreases with decreasing the sediment supply (Figs. 5 and 11). The decrease in bar height weakens the flow meander and can increase the bar wavelength. In Figures 11 and 13 of Chatanantavet and Parker (2018), the bar width increases as the alluvial cover increases, but the wavelength does not increase (almost constant). Both results do not support your assumption.

You may be confusing the length of an individual bar patch with the length between two bar patches (i.e., wavelength shown in Fig. 2 in your paper). Although the bar patch length has a positive correlation with both the bar width and the alluvial cover, the wavelength has no positive correlation (negative in numerical analysis, constant in experiment). I think $\gamma$ depends on the bar wavelength or bar height, and decreases as alluvial cover decreases.

I encourage you to reconsider your assumption and model.

---

## Referee Comment (RC3) · Anonymous Referee #2 · 2 Dec 2019

**Review for manuscript "Mass balance, grade, and adjustment timescales in bedrock channels" by Turowski, submitted to ESurf**

In this paper, the author investigated the adjustment timescales of width, slope, and bed cover for bedrock rivers, via theoretical framework and numerical computations. I think the idea is significant and interesting, especially he included the lateral erosion. The English is very good. I just have some comments below, one of which may affect the orders of magnitude of adjustment timescale, however. So please consider.

Seeing the exchanges between the anonymous reviewer #1 and the author, I went back and checked the paper by Chatanantavet and Parker 2008. In their figure 11 (especially comparing the subfigures 2 and 4), at first glance I thought the assumption by the author Turowski was correct, i.e. the bar wavelength decreases with decreasing fraction of alluvial cover. But I could be wrong since I have not done any direct research regarding alternate bars or meandering channels. In the last interactive comment, the anonymous reviewer #1 stated that "although the bar patch length has a positive correlation with both the bar width and the alluvial cover, the wavelength has no positive correlation." It is hard to assess quantitatively and would need a longer flume length. I leave it to the AE and the editor to digest.

**Major comments:**

- I really think you should include a factor of "flood intermittency" (a fraction of time duration in a year that has water discharge significant enough to do the majority of bedrock abrasion). This is commonly done in any morphodynamical modeling of such a temporal process involving high flow: see any papers done by the research groups of Chris Paola and Gary Parker (e.g., Chatanantavet and Parker 2009). For example, your Exner equation (eq.4 and then eq. 28, 34) does not have this factor and go on to derive the timescale for slope adjustment (eq 37). Say, if flood intermittency is equal to 0.05-0.1 in a particular location. Then your slope adjustment timescale could be missed by a factor of 10-20. That is significant and may affect your conclusion. I think it would make the adjustment timescale longer. In table 1, for example, you wouldn't expect that water discharge of 40 m3/s is present for the entire year in the Liwu river.

- The part where you talk about lateral erosion and alternate bars (i.e. section 2.2 and elsewhere); I think that it is worth or even very important to note to the readers that these morphological configurations occur only in a specific range of channel slope in natural setting, which is around S = about 0.1-3% per Montgomery and Buffington 1997, and other studies. Beyond this slope range, i.e. at S = 3% or higher until S = about 10%, steep-pool configuration dominates bedrock channels and its associated sediment transport differ quite significantly since there is strong coupling interaction between hydraulic jump hydrodynamics and sediment trajectory/movement (see any flume experimental work in step pool). Hence, in your paper

when you talk about lateral erosion and alternate bars, the conceptual model may be limited to slope of no more than 2-3% (or 0.02-0.03). Seeing that slope in your results span until 0.1 (figure 5D), it is a bit farfetched. This slope cutoff is eminent whenever I conducted flume experiments ranging slope from 0.1% to 5%; once the slope hit 3% the step pools were very obvious and the hydraulics and associated sediment transport were so much different from alternate bars (or pool riffle) or plane-bed feature.

- P12 L5; critical Shields stress also varies with channel slope (e.g., Lamb et al, 2008, JGR-ES; Chatanantavet et al. 2013, JGR ES). I know traditionally and simplistically people assume that it is constant, but it is an old concept. And this can affect your numerical results greatly because unlike alluvial rivers, bedrock rivers has varying slopes in a high value range (around 0.001-0.1, in which alluvial rivers don't touch but odd things happen here such as hydraulic jumps).

- Page 10; the response time ratios. Sorry, I don't get why you wrote up this section. I don't see its usefulness and you didn't explain why this needs to be done. You also did not use any of these to plot the results or discuss about it.

- There is a paper by Sklar and Dietrich 2006 (Geomorphology) titled "The role of sediment in controlling steady-state bedrock channel slope: Implications of the saltation–abrasion incision model". I think it is worth to check it out if you have not already. Actually their work is highly related to yours, along the same concept (i.e. their figure 6 vs your figure 4) and should be acknowledged. I understand that your work added lateral erosion and so on, which is cool. Actually looking at their figure 6, it reminds me that sensitivity analysis should be implemented with this kind of studies.

- If I understand correctly, your results in figures 4, 5, 6 are dealing with specific boundary conditions at any specific point/reach section in a channel. But I am afraid, as the figures stand now, the presentation might mislead some readers to think that slope and channel width (and cover) are spatially constant along a whole bedrock channel length. As you know, both slope and channel width are not spatially constant along bedrock channels. And we often see concave or convex or straight bedrock streams. When investigating steady state conditions of river channels, I think it would be cool to see plots of spatially distributed features of the variables in questions. OR at least **discuss** about it, or even mathematically. This is especially when you show "reach length" of 10 km in Table 1. So the readers may visualize and think you are talking about the whole channel length. I feel like the work is incomplete by having no spatially distributed results or talking/discussing about it. You have great math framework already and some initial results in figs 4-6. Having these additional figures would enhance the paper nicely (in that case, you might need to add some equations to implement).

**Minor comments**

P1, L9; an alluvial (use lower case after colon)

P1, L11, 14; "…a balance between channel incision and uplift" sounds better, I think.

P1, L13; I think "in the present work" sounds more formal and commonly used than "within the present paper"

P1, L19; if these are from your results, please indicate clearly by saying "My results show that …" or something like that.

P1 L29; various timescales

P1 L35; delete "for"

P1 L38; is temporally constant

P3 L6-L18; in this paragraph, I think you should explicitly state somewhere that you only investigate the bedrock incision process due to bedload abrasion, and NOT consider plucking, suspended abrasion, etc. Also in discussion section, you don't touch this topic.

P3 L1-L4; you may want to add a reference here such as Chatanantavet and Parker 2009 and/or a few other studies who used this equation to show how bedrock rivers approach a steady state. Readers who wish to read further in details can see how steady state profiles look like for bedrock channels.

P3 L23; this sentence is quite awkward. Consider reword.

P5; you have here 2.2.1 but then 2.3 . I think probably you better just delete sub-section 2.2.1 and merge the text with 2.2.

P10 L15-16; the font size here is different.

---

## Author Comment (AC2) · 16 Jan 2020

Dear Editors, dear reviewers,

Many thanks for the comments on the paper. I have addressed everything to my best abilities and think that the paper has improved because of it. I hope there are no further queries.

Below, I reply to the reviewers' comments in detail, given first their comment, then my reply in *italics*.

With best wishes, Jens Turowski

Summary of changes

- I removed previous figure 5. The calculations shown there were unrealistic and only of technical interest. Since they seem to have confused Reviewer #3, I have decided to remove them from the paper. This does not affect the central argument.
- I have added a paragraph in the discussion (section 4.1), discussing potential limits of the model assumptions. In particular, the issue of the bar wavelength and scaling is discussed in some detail.
- I have gone through the text, trying to improve flow, clarity and readability.

Reviewer #1

In this manuscript, the author proposed a mechanistic model for analyzing the adjustment timescales for channel width, channel bed slope and alluvial bed cover in a mixed alluvial - bedrock channel. However, in the current version a significant question on the assumption of the bar wavelength remains and needs to be addressed.

In eq. 23, you assumed that the bar wavelength decreases with decreasing the fraction of alluvial cover. However, recent studies indicate that the bar wavelength increases with decreasing the fraction of alluvial cover in mixed alluvial – bedrock channel, in theoretically (Nelson and Seminara, 2011, Fig.2b, doi: 10.1029/2011GL050806) and numerically (Inoue et al., 2016, Figs 5 and 11, doi: 10.1061/(ASCE)HY.1943-7900.0001124.). Experiments conducted by Chatanantavet and Parker (2008) also show no decrease in bar wavelength. Your assumption is based on Kelly (2006)'s observations in alluvial channel, but may not be applicable to mixed alluvial – bedrock channel. Because this assumption directly affects the lateral erosion rate and the timescales, the results shown in Figs 4 – 6 may be incorrect.

*I acknowledge that the assumption I have made is based on data for alluvial streams. I made this assumption because there is little (no) relevant data available for bedrock channels. This was stated in the original manuscript. The reviewer disputes this statement, citing three articles for support, two modelling papers (Nelson and Seminar, 2011, which was likely confused with Nelson and Seminara, 2012; Inoue et al., 2016) and one experimental paper (Chatanantavet and Parker, 2008). As I have already stated in my initial reply in the discussion forum, I was not able to find this evidence in the mentioned papers. In his reply to my comment in the forum, the reviewer mentions another paper by Chatanantavet and Parker, 2018. Below, I will comment on all of these papers, and elaborate my point of view on this in a little more detail.*

*First, none of the mentioned papers was set up to investigate the problem of bar length, bar geometry and bar wavelength. The reviewer has not been able to point out explicit relevant*

*statements on the matter within the mentioned papers, and instead cites various figures, especially from the experimental paper, in support.*

*Nelson and Seminara, 2011: This paper deals with channel cross-sectional shape and does not mention bars.*

*Nelson and Seminara, 2012: Here, the authors investigate initial bar instability, not bar geometry. They explicitly state that their analysis is not suitable for making statements about the emerging forms (paragraph 25):* "It is important to emphasize that the linear analysis presented here only addresses the initial instability which generates bar like patterns. Predicting the actual pattern emerging from this process will require a fully nonlinear analysis possibly able to treat regions where the areal sediment concentration C locally reaches 1 and local alluviation occurs." *It would be the latter (steady state bar geometry) that is relevant for my model. Concluding, the Nelson and Seminara 2012 paper does not contain statements relevant for the debate.*

*Inoue et al., 2016: The authors use a numerical model to study the transient adjustment of cover and bedforms, keeping boundary conditions constant. For this paper, the reviewer refers to Figures 5 and 11 in his argument. Figure 5 shows 6 maps at consecutive times, and indeed, here it looks like bar wavelength is constant as deposition continues. Figure 11 shows three similar time slices. Alternating bars appear in the third (last; 500 hours) shown slice, and a comparison of bar wavelength for different slices is thus not possible. If the deposition in time slice 2 (250 hours) is interpreted to show alternating bars, the wavelength seems to be longer, of the order of the length of the experimental reach. In this interpretation, Figure 7 would suggest an evolution of bar wavelength over the course of the experiment. There is another relevant figure in the paper, Figure 14, which shows three time slices of a simulations set to correspond to conditions studied by Chatanantavet and Parker, 2009, in experiments. These can also be interpreted to show bar wavelength that is not changing over the course of the experiment. There is a fundamental difference between the model conditions studied in this paper and the assumptions I make for my model set up: While Inoue et al. study transient adjustment to a steady state cover, starting from an empty bed, all applications within my paper build on the assumption of steady state cover (eq. 32; see also Turowski and Hodge, 2017). A comparison also needs to take into account this aspect.*

*Regarding both modelling papers (Nelson and Seminara, 2012; Inoue et al., 2016), I would like to also repeat the statements from Inoue et al., that I quoted in my comment in the forum (page 8, left-hand column, 2nd full paragraph):*
"Nelson and Seminara (2012) conducted a linear stability analysis of bars on the bedrock and analyzed the wavelength of infinitesimal bars. The findings of their analysis are as follows: (1) regions where alternate bars form on the bedrock are determined not only by the width/depth ratio, but to some degree by the ratio of τ =τc; and (2) the wavelength of the bars increases with decreasing sediment supply rate. The analysis by Nelson and Seminara (2012), unlike the simulation of this study, did not consider localized bedrock erosion by bedload; therefore, it is not possible to compare the two simulations quantitatively. However, the two models show a similar tendency to form longer wavelength bars when the sediment supply is lower." *I read this to support my assumption. In his/her reply to the comment, the reviewer did not explicitly address this quote. She/he did state, however, that* "You may be confusing the length of an individual bar patch with the length between two bar patches". *This may be the case, but given the sparsity of information it seems to be the most straightforward interpretation of the above statement.*

*Chatanantavet and Parker, 2008: The authors use flume experiments to investigate how cover changes with sediment supply (or rather, the ratio between supply and transport capacity) for various conditions and bed topographies. They mention that alternating bars were present in the*

*experiments (e.g., paragraph 13), but do not give details on their morphology or how their wavelength scales with cover. Figure 11 seems to be the only figure containing relevant material. The question of bar wavelength is difficult to assess from this figure: picture quality is low because of water surface reflections, there is a single wavelength within the shown part of the flume, and it is unclear whether this shows a transient or steady state. As evidence, this is at best suggestive or circumstantial. Reviewer #3 agrees on this assessment and states explicitly that she/he interprets this figure to support my assumption, rather than the claim of reviewer #1. I hesitate to make a final judgement on such thin evidence.*

*Chatanantavet and Parker, 2018: I was not able to find this paper. Please supply a full reference.*

*In summary, the evidence presented in the three mentioned papers is at most suggestive. There is an additional complication. Even if I was convinced that bar wavelength is independent of cover, bar wavelength needs to depend on something. It seems safe to me to state that we currently do not understand the geometry of alternating bars in bedrock channels and what controls it. Simple dimensional analysis suggests that at least one other length scale is required. I chose bar width for this length scale, keeping the aspect ratio constant. As long as we do not have full understanding of the controls, another assumption needs to be made (for example, a dependence on channel width or flow depth), for which there is little evidence either. In light of the currently available evidence, my strategy of using an alluvial analogue seems to me still the best and most plausible option. I would be very happy to change this approach if convincing evidence is supplied.*

*All this said, I repeat my statement from the reply comment in the forum: A change in the bar geometry affects only the lateral erosion equation (eq. 24 in the paper). This propagates into the response time of width for a widening channel, but not the response time of cover, of bed slope or for the width for a narrowing channel. The response times for widening will be substantially affected only when the bar aspect ratio deviates substantially from the value of 2-10 that I assumed (5 for the example calculations). This is likely the case only for low values of bed cover. Changing the assumption on bar geometry does not affect the steady state channel morphology presented in Fig. 4. In summary, a change in the dependence of bar wavelength on cover would not change the arguments and conclusions of the paper. The issue of bar wavelength is a minor part in the argument of the paper and does not change the overall conclusions, the narrative and the general points that I am trying to make. As a result, I find the overall negative assessment of the paper, based on this single minor criticism, to be unjustified.*

*In response to the reviewer's comment, I added a paragraph in the discussion on the bar geometry issue. I also point out the caveat mentioned in the forum comment that due to the constant aspect ratio, bar wavelength approaches zero as cover approaches zero. This seems to be unphysical and needs to be addressed in a fully dynamic model. I also stress that for all the calculations presented in this paper, the assumption of steady state cover is made.*

Additional comments by line number below:
P2 Eq. (1): The density of the sediment?
*Changed.*
P7 Line 9: Auel et al., 2017 a or b?
*2017a, corrected.*
P7 Line 15: Why does the secondary flow not affect the lateral impact velocity?
*It probably does, but we have few constraints on it. The available data suggest that roughness is the most important control. See the discussion in Turowski, 2018.*

P7 Line 32: Nelson and Seminara analyzed alternate bars in a mixed alluvial – bedrock channel in 2012, not 2011. The paper listed in the reference is probably incorrect.
*Corrected.*
P12 Figure 4c: There is no explanation of Fig4c in the text. Why does slope and width change with uplift rate?
*The assumption here was that incision rate is equal to uplift rate in steady state.*
P15. Line 6: Gravel bars do not increase their wavelength as cover increases.
*See discussion above on the major point.*
P19 Eq. (B6): When C is close to 0 (i.e., almost completely exposed bedrock), is close to 90 degree (i.e., sediments move towards the sidewalls). Why?
*This is due to the coupling of bar wavelength to cover – the amplitude of the sine wave is small in comparison to the channel width, making the angle very steep. This seems unphysical. I have added a paragraph in the discussion.*
P26-P29: Inoue et al. (2014), Montgomery et al. (1996), Shepherd (1972) and Whipple (2004) are not listed in the references.
*Missing references added.*

Reviewer #3
In this paper, the author investigated the adjustment timescales of width, slope, and bed cover for bedrock rivers, via theoretical framework and numerical computations. I think the idea is significant and interesting, especially he included the lateral erosion. The English is very good. I just have some comments below, one of which may affect the orders of magnitude of adjustment timescale, however. So please consider.

Seeing the exchanges between the anonymous reviewer #1 and the author, I went back and checked the paper by Chatanantavet and Parker 2008. In their figure 11 (especially comparing the subfigures 2 and 4), at first glance I thought the assumption by the author Turowski was correct, i.e. the bar wavelength decreases with decreasing fraction of alluvial cover. But I could be wrong since I have not done any direct research regarding alternate bars or meandering channels. In the last interactive comment, the anonymous reviewer #1 stated that "although the bar patch length has a positive correlation with both the bar width and the alluvial cover, the wavelength has no positive correlation." It is hard to assess quantitatively and would need a longer flume length. I leave it to the AE and the editor to digest.

Major comments:
- I really think you should include a factor of "flood intermittency" (a fraction of time duration in a year that has water discharge significant enough to do the majority of bedrock abrasion). This is commonly done in any morphodynamical modeling of such a temporal process involving high flow: see any papers done by the research groups of Chris Paola and Gary Parker (e.g., Chatanantavet and Parker 2009). For example, your Exner equation (eq.4 and then eq. 28, 34) does not have this factor and go on to derive the timescale for slope adjustment (eq 37). Say, if flood intermittency is equal to 0.05-0.1 in a particular location. Then your slope adjustment timescale could be missed by a factor of 10-20. That is significant and may affect your conclusion. I think it would make the adjustment timescale longer. In table 1, for example, you wouldn't expect that water discharge of 40 m3/s is present for the entire year in the Liwu river.
*This is an excellent point. The representative discharge I used for the calculation is actually a representative discharge of all flows that transport bedload and could therefore contribute to erosion. The method for the discharge partitioning was developed by Sklar and Dietrich (2006). This is described in*

*the Turowski et al. (2007) paper, from which the numbers originate, but was not explained in the present manuscript. I have now added this explanation. As such, flood intermittency has been taken into account in an implicit way.*

- The part where you talk about lateral erosion and alternate bars (i.e. section 2.2 and elsewhere); I think that it is worth or even very important to note to the readers that these morphological configurations occur only in a specific range of channel slope in natural setting, which is around S = about 0.1-3% per Montgomery and Buffington 1997, and other studies. Beyond this slope range, i.e. at S = 3% or higher until S = about 10%, steep-pool configuration dominates bedrock channels and its associated sediment transport differ quite significantly since there is strong coupling interaction between hydraulic jump hydrodynamics and sediment trajectory/movement (see any flume experimental work in step pool). Hence, in your paper when you talk about lateral erosion and alternate bars, the conceptual model may be limited to slope of no more than 2-3% (or 0.02-0.03). Seeing that slope in your results span until 0.1 (figure 5D), it is a bit farfetched. This slope cutoff is eminent whenever I conducted flume experiments ranging slope from 0.1% to 5%; once the slope hit 3% the step pools were very obvious and the hydraulics and associated sediment transport were so much different from alternate bars (or pool riffle) or plane-bed feature.
*Excellent point. I have added a paragraph on the limits of model assumptions to the discussion in 4.1, mentioning this and some other points. I left the presentation of results as is for the interest of the reader. Model assumptions are clearly laid out and there should hopefully be no confusion for a careful reader.*

- P12 L5; critical Shields stress also varies with channel slope (e.g., Lamb et al, 2008, JGR-ES; Chatanantavet et al. 2013, JGR ES). I know traditionally and simplistically people assume that it is constant, but it is an old concept. And this can affect your numerical results greatly because unlike alluvial rivers, bedrock rivers has varying slopes in a high value range (around 0.001-0.1, in which alluvial rivers don't touch but odd things happen here such as hydraulic jumps).
*I am of course aware of this. I have consciously decided not to use the Lamb equation – first, it is unphysical in the limit of low slopes, where the value should stabilize around a classical value of 0.045 or so. There is also a temporal dependence complicating the picture (see for example recent work of Claire Masteller). And the explicit dependence of slope would add yet another feedback to already complicated equations. There are also field data, refuting the simple trends described by the Lamb equation (see for example publications by Kristin Bunte). In the end, the addition would not majorly change trends, it would not yield any further interesting insights, and would not change the main argument and message of the paper.*

- Page 10; the response time ratios. Sorry, I don't get why you wrote up this section. I don't see its usefulness and you didn't explain why this needs to be done. You also did not use any of these to plot the results or discuss about it.
*The response time ratios are plotted in Figures 5 b, d, and f (previously Fig. 6). The argument put forward in section 4.3 is based on these calculations. I added a couple explanatory sentences to the start of section 2.3.4.*

- There is a paper by Sklar and Dietrich 2006 (Geomorphology) titled "The role of sediment in controlling steady-state bedrock channel slope: Implications of the saltation–abrasion incision model". I think it is

worth to check it out if you have not already. Actually their work is highly related to yours, along the same concept (i.e. their figure 6 vs your figure 4) and should be acknowledged. I understand that your work added lateral erosion and so on, which is cool. Actually looking at their figure 6, it reminds me that sensitivity analysis should be implemented with this kind of studies.

*I am aware of this paper, but had not read it for some time. I do not want to give a full criticism of this paper here, but I think the linear decomposition (their eq. 32) is incorrect, and for this reason their results are fundamentally flawed.*

*I am not quite sure what the reviewer is asking for here. The equivalent to their Fig. 6 is my Fig. 4c. A cross-comparison of different model approaches is beyond the scope of my paper, and in my mind not particularly useful, because sufficient data for a clean discrimination are currently lacking. In any case, I have already demonstrated in a previous paper (Turowski ESurf 2018) that the model predictions for steady state are in agreement with all currently available data on the reach scale, because it predicts the observed scalings of width and sinuosity in addition to that of slope. This is in contrast to any other models I know of. With regard to steady state geometry, the novelty of the present paper is the quantification of the sideward deflection length scale d. Figure 4 demonstrates that this quantification does not change the analysis made in the previous paper. This point is made and discussed in section 4.2, where I have now added a sentence to make this clearer.*

*In the revised manuscript, the Sklar and Dietrich 2006 paper is now cited because it describes a discharge partitioning method used to obtain the representative discharge for the Liwu (Table 1).*

- If I understand correctly, your results in figures 4, 5, 6 are dealing with specific boundary conditions at any specific point/reach section in a channel. But I am afraid, as the figures stand now, the presentation might mislead some readers to think that slope and channel width (and cover) are spatially constant along a whole bedrock channel length. As you know, both slope and channel width are not spatially constant along bedrock channels. And we often see concave or convex or straight bedrock streams. When investigating steady state conditions of river channels, I think it would be cool to see plots of spatially distributed features of the variables in questions. OR at least discuss about it, or even mathematically. This is especially when you show "reach length" of 10 km in Table 1. So the readers may visualize and think you are talking about the whole channel length. I feel like the work is incomplete by having no spatially distributed results or talking/discussing about it. You have great math framework already and some initial results in figs 4-6. Having these additional figures would enhance the paper nicely (in that case, you might need to add some equations to implement).

*I do not fully understand this comment.*

*In Figure 4, I show steady state slope and width as a function of forcing parameters (uplift rate, water discharge, sediment supply). Here, the dependence on slope and width can be explicitly seen – and they are mostly not constant! Note for example the concavity of the channel in Fig. 4a – the decline only looks linear because of the log-log scale. The interesting exception is that width is predicted to be explicitly independent of water discharge. This is surprising because we all know about the typical scaling W~sqrt(Q). This scaling arises in the model from the covariance of water and sediment discharge. The point is discussed in some detail in section 4.2. See also the discussion in Turowski, ESurf 2018.*

*Figures 5 and 6 show response time scales, rather than channel geometry. For Figure 5, I used the values from Table 1, for Figure 6, slope, width and cover were calculated according to the model. I do not see how these could give the impression of constant slope, width, or cover. The value of the reach length is needed for these calculations, because slope is adjusted by knickpoints migration, which needs to move through the entire reach for a full adjustment. Similarly for the adjustment of cover: for a given supply*

*rate, adjustment times obviously are dependent on the amount of area that needs to be covered, which is set by the product of length and width.*

*A reach is defined as a stretch of the river over which boundary conditions and, as a consequence, channel geometry is roughly constant. So it should not come as a surprise that width and slope are constant within a reach.*

*The reviewer asks for a plot of 'spatially distributed features'. I understand this as a plot against river length or some kind of other distance. But, in essence, the plot against discharge (Fig. 4a) is doing exactly that. River length is not a control variable. To produce such a plot, I would need to make an assumption about how discharge scales with drainage area (hydrology, for example Q~P\*A, where P is the precipitation rate), and then an assumption about how drainage area scales with length (basin geometry, for example Hack's law). These assumptions may apply in some regions but not in others. Plotting against discharge is more natural, as it keeps the relationship between forcing and response explicit and direct. I do not think such a plot would be useful and have not included one.*

*In summary, I think that Fig. 4 is essentially supplying the information that the reviewer is asking for. There seems to be some misunderstanding about the contents of Fig. 5 and 6, but I am unsure about what that is exactly.*

*I have tried to improve the clarity of the text. I have also removed Fig. 5 to avoid confusion. The information in this figure was mainly of technical interest and can be easily reproduced with the information given in the paper.*

**Minor comments**

P1, L9; an alluvial (use lower case after colon)

*Changed.*

P1, L11, 14; "…a balance between channel incision and uplift" sounds better, I think.

*Changed.*

P1, L13; I think "in the present work" sounds more formal and commonly used than "within the present paper"

*Changed.*

P1, L19; if these are from your results, please indicate clearly by saying "My results show that …" or something like that.

P1 L29; various timescales

*Corrected.*

P1 L35; delete "for"

*Changed to 'in'.*

P1 L38; is temporally constant

*I prefer the current phrasing. No changes.*

P3 L6-L18; in this paragraph, I think you should explicitly state somewhere that you only investigate the bedrock incision process due to bedload abrasion, and NOT consider plucking, suspended abrasion, etc. Also in discussion section, you don't touch this topic.

*This is a good point; however, this is not the right paragraph, because mechanisms of erosion have not been introduced here. I added a statement at the beginning of section 2.*

P3 L1-L4; you may want to add a reference here such as Chatanantavet and Parker 2009 and/or a few other studies who used this equation to show how bedrock rivers approach a steady state. Readers who wish to read further in details can see how steady state profiles look like for bedrock channels.

*The shape of a bedrock channel long profile depends on the assumptions of the erosion mechanisms, and its mathematical description. Eq. 1 has been used in many studies – most current landscape evolution models use it as a basic mass balance equation, it is also used for stream-profile inversion using the*

*stream power model. I think citations to particular modelling studies would be misleading here. I could not find many papers explicitly stating the equation – for example, the early Whipple and Tucker papers always give the stream power model first and then state 'combined with a statement of conservation of mass' or similar. I have added a citation of Howard 1994, who explicitly stated the equation.*

P3 L23; this sentence is quite awkward. Consider reword.

*Moved 'third' to the start of the sentence.*

P5; you have here 2.2.1 but then 2.3 . I think probably you better just delete sub-section 2.2.1 and merge the text with 2.2.

*Removed the sub-heading to 2.2.1.*

P10 L15-16; the font size here is different.

*Changed.*

---

## Author Response (AR2)

Minor language changes:

I have done a few minor language changes in the production files in comparison to the review version of the paper. These were mainly to fix typos and small grammatical errors.

I have added a sentence in the end of the 2nd paragraph of section 4.1 (in italics, quote longer for context) for clarification: "Second, at high cover values, neighbouring bars start to overlap and the relationship to cover likely becomes more complicated. *For a fully covered bed, bar dynamics should resemble those in alluvial channels.*"